# Long Non-Coding RNA in the Pathogenesis of Cancers

**DOI:** 10.3390/cells8091015

**Published:** 2019-09-01

**Authors:** Yujing Chi, Di Wang, Junpei Wang, Weidong Yu, Jichun Yang

**Affiliations:** 1Department of Central Laboratory & Institute of Clinical Molecular Biology, Peking University People’s Hospital, Beijing 100044, China; 2Department of Physiology and Pathophysiology, School of Basic Medical Sciences, Peking University Health Science Center, Beijing 100191, China; 3Key Laboratory of Cardiovascular Science of the Ministry of Education, Center for Non-coding RNA Medicine, Beijing 100191, China

**Keywords:** lncRNAs, cancer, proliferation, metastasis

## Abstract

The incidence and mortality rate of cancer has been quickly increasing in the past decades. At present, cancer has become the leading cause of death worldwide. Most of the cancers cannot be effectively diagnosed at the early stage. Although there are multiple therapeutic treatments, including surgery, radiotherapy, chemotherapy, and targeted drugs, their effectiveness is still limited. The overall survival rate of malignant cancers is still low. It is necessary to further study the mechanisms for malignant cancers, and explore new biomarkers and targets that are more sensitive and effective for early diagnosis, treatment, and prognosis of cancers than traditional biomarkers and methods. Long non-coding RNAs (lncRNAs) are a class of RNA transcripts with a length greater than 200 nucleotides. Generally, lncRNAs are not capable of encoding proteins or peptides. LncRNAs exert diverse biological functions by regulating gene expressions and functions at transcriptional, translational, and post-translational levels. In the past decade, it has been demonstrated that the dysregulated lncRNA profile is widely involved in the pathogenesis of many diseases, including cancer, metabolic disorders, and cardiovascular diseases. In particular, lncRNAs have been revealed to play an important role in tumor growth and metastasis. Many lncRNAs have been shown to be potential biomarkers and targets for the diagnosis and treatment of cancers. This review aims to briefly discuss the latest findings regarding the roles and mechanisms of some important lncRNAs in the pathogenesis of certain malignant cancers, including lung, breast, liver, and colorectal cancers, as well as hematological malignancies and neuroblastoma.

## 1. Introduction

It was estimated that there were about 18.1 million newly diagnosed cancer cases and about 9.6 million cancer-related deaths worldwide in 2018 [1]. Lung cancer (LC) has the highest incidence and mortality rate among human cancers. Female breast cancer (BC), prostate cancer (PCA), and colorectal cancer (CRC) are the second, third, and fourth cancers with the highest incidence, respectively. CRC, gastric cancer (GC), and hepatocellular carcinoma (HCC) are the three cancers with the highest mortality rate beyond LC [1]. The main risk factors that influence the incidence and mortality of cancers include rapid population growth and aging, socioeconomic development and patient’s low screening compliance caused by lower education and income, and lack of health insurance and awareness [2]. So far, most cancers are not effectively diagnosed at the early stage. At present, common cancer treatments include surgery, chemotherapy, radiation therapy, laser therapy, and combination therapy [3]. Because of the limited and unspecific serum cancer biomarkers for advanced-stage diagnosis [4] and cancer-related drug resistance, the therapeutic effects for invasion-related and metastasis-related cancers are still very poor [3]. Therefore, it is urgent to find novel biomarkers and targets that are more effective for the early diagnosis, treatment, and prognosis of cancers than traditional methods and targets.

Long non-coding RNAs (lncRNAs) are a class of RNA transcripts with a length larger than 200 nucleotides. Generally, lncRNAs do not encode proteins or peptides. In addition to the size of other classes of non-coding RNAs (microRNAs, small interfering RNAs, small nucleolar/nuclear RNAs), lncRNAs also have secondary and three-dimentional structures which enable them to have both RNA- and protein-like functions [5]. LncRNAs can be predicted using several online prediction tools based on Coding Potential Calculator algorithm version 2 (CPC2) freely at http://cpc2.cbi.pku.edu.cn [6], and can also be predicted using software such as CNCI (http://www.bioinfo.org/software/cnci), CPAT (http://lilab.research.bcm.edu/cpat/index.php), ESTScan (http://estscan.sourceforge.net/), PLEK (https://sourceforge.net/projects/plek/files/), PORTRAIT (http://bioinformatics.cenargen.embrapa.br/portrait), FEELnc (https://github.com/tderrien/FEELnc), TransDecoder (http://trinityrnaseq.sf.net), and GeneMarkS-T (http://topaz.gatech.edu/GeneMark/license_download.cgi); CPAT and ESTScan can also provide a web server [7]. Moreover, recently, we developed another effective method—which is named Gene Importance Calculator (GIC)—for predicting the essentiality of lncRNAs with high accuracy and sensitivity (http://www.cuilab.cn/gic/) [8].

It has been shown that the majority of lncRNAs are localized in the nucleus [9], but some of the lncRNAs also play roles in cytoplasm [10]. Moreover, some lncRNAs can be transmitted to adjacent cells or serum through exosome trafficking [11]. LncRNAs regulate the target gene expression, mainly through *cis*-regulation or *trans*-regulation [12]. It has been estimated that there are more than 60,000 lncRNAs in humans, and the number of lncRNAs is still increasing quickly [13]. So far, the functions of only a very few number of lncRNAs have been annotated [14,15], and various methodologies have been developed to explore the expression, distribution, and function of lncRNAs (Table 1). By using bioinformatic and high throughout methods, recent studies have revealed that the dysregulated lncRNA profile is widely involved in the pathogenesis of tumors, which includes cell proliferation, migration, invasion, epithelial-to-mesenchymal transition (EMT), apoptosis, and anti-tumor drug resistance [16,17,18,19,20]. These findings suggest that some lncRNAs are potential targets and biomarkers for the diagnosis and prognosis of malignant tumors. In this review, the roles and mechanisms of some important lncRNAs in the pathogenesis of lung, breast, liver, and colorectal cancers, as well as hematological malignancies and neuroblastoma, are briefly discussed.

## 2. LncRNAs Definition and Functions

LncRNAs are a class of RNA transcripts with a length greater than 200 bp, characterized by more spatial and temporal specificity and lower interspecific conservation when compared with mRNAs [21]. LncRNAs had been previously considered to be the by-products of the transcription process. However, it has been widely accepted now that lncRNAs are involved in the process of cell differentiation and growth and the pathogenesis of many diseases, including cancer [22]. According to the relative location of lncRNAs to protein-encoding genes in the genome, they have been classified into sense lncRNA, antisense lncRNA, bidirectional lncRNA, intron lncRNA, intergenic lncRNA, and enhancer lncRNA [23]. So far, lncRNAs have been revealed to play an important role in regulating gene expression at the epigenetic, transcription, and post-transcriptional levels [24].

Firstly, lncRNAs play a crucial role in regulating gene expression though chromatin modification and remodeling, histone modification, and nucleosome localization changes [25]. Among the mechanisms regulating chromosome remodeling, SWItch/Sucrose Non-Fermentable (SWI/SNF) complex is a chromosome reconstitution complex composed of multiple subunits, which drive the change of nucleosome localization [26,27]. LncRNAs can interact with SWI/SNF complexes to alter chromosome structure and regulate gene expression [28,29,30,31]. In addition, lncRNAs also regulate chromatin remodeling by affecting the DNA methylation with S-adenosylhomocysteine hydrolase [32], DNA demethylation with growth arrest and DNA-damage-inducible alpha (GADD45A) [33], and acetylation of histone with Sirtuin 6 (SIRT6) [34]. Histone is an important basic unit of the nucleosome, which plays an important role in the formation of structure and function of chromosomes. Many lncRNAs modulate histone methylation by interacting with polycomb repressive complex 2 (PRC2), altering the structure and function of chromosomes [35,36].

Furthermore, microRNAs (miRNAs) inhibit the translation efficiency and/or induce mRNA degradation of the target gene by complementary pairing with the target RNA, regulating gene expression at the post-transcriptional level. There is a type of competitive endogenous RNA (ceRNA) that can indirectly regulate the expression of target genes via competitively binding with miRNAs; this competitive binding of miRNAs is also called microRNA sponges [37,38]. Many lncRNAs can work as miRNA sponges to influence gene expression and the development of cancer [39].

## 3. LncRNAs in Lung Cancer (LC)

LC is the leading cause of cancer-related deaths, accounting for 18.4% of total cancer-related deaths and taking the first place with an incidence rate of 11.6% among males and females [1]. It was estimated that there were 2,093,876 newly diagnosed LC cases and 1,761,007 cases that died from LC worldwide in 2018 [1,40]. LC is pathologically divided into small cell lung cancer (SCLC) and non-small cell lung cancer (NSCLC), which accounts for 80–85% of all LC cases. NSCLC is further divided into lung squamous cell carcinoma (LSCC), lung adenocarcinoma (LAD), and large cell carcinoma (LCC) [41,42]. Although many advances had been made in the treatment of LC, such as surgical resection and chemotherapy in the past decades, the five-year survival rate is only 15% [41] due to its late-stage diagnosis, metastasis, insensitivity to chemotherapy, and recurrence [41,43]. Clearly, to explore new targets and biomarkers for early diagnosis and more effective treatment of LC is of great significance. In particular, several lncRNAs have been shown to have the potential of serving as novel biomarkers and targets for LC diagnosis and treatment [4].

### 3.1. Metastasis-Associated Lung Adenocarcinoma Transcript 1 (MALAT1)

MALAT1, also known as nuclear-enriched abundant transcript 2 (NEAT2), is an 8.7 kb intergenic lncRNA located on chromosome 11q13 [4], and is one of the most abundantly expressed lncRNAs in normal tissues [44]. MALAT1 is involved in post-transcriptional regulation of gene expression and mRNA splicing [45]. It has been reported that MALAT1 is involved in the pathogenesis of LC and other human cancers, including liver, breast, colon, uterus, prostate, ovarian, and hematological malignancies and neuroblastoma [41,46,47,48].

Several studies have suggested that MALAT1 could serve as a potential prognostic biomarker and therapeutic target for patients with early-stage LC due to its specificity and stability [41,44,49,50,51]. It has been shown that the expression of MALAT1 in NSCLC tissues is higher than that in adjacent tissues, and NSCLC patients with high MALAT1 expression have poor overall survival (OS) [50,52,53]. Moreover, the expression of MALAT1 in NSCLC tissues with bone metastasis is higher than that in NSCLC tissues without bone metastasis [54]. MALAT1 expression level is lower in the peripheral whole blood [55] and cellular fraction of human blood [56] in patients with LC than that in healthy subjects. However, the expression of MALAT1 in the whole blood of LC patients with metastasis is shown to be higher compared with that in patients with non-metastasis [55]. Moreover, MALAT1 level is significantly upregulated in the whole blood of LC patients with metastasis such as bone or brain metastasis than that in LC patients with lymph node or pleura metastasis [55].

Functionally, MALAT1 is considered as an oncogenic lncRNA because of its role in promoting tumor differentiation, proliferation, migration, invasion, EMT, and chemoresistance [4,41]. Silencing of MALAT1 inhibits cell proliferation and colony formation in NSCLC cell lines [49,57,58]. MALAT1 promotes cell migration and invasion process by sponging miR-206 in A549 and H1299 cell lines [59]. In addition, MALAT1 can promote EMT and invasion of A549 and H1299 cells by upregulating the expression of snail family transcriptional repressor 2 (SLUG) through competitively sponging miR-204 [60]. C-X-C motif chemokine ligand 5 (CXCL5) is a downstream gene of MALAT1, and knockdown of CXCL5 in vitro could inhibit MALAT1-induced cell migration and invasion [52]. In nude mice, the total number and area of lung tumor nodules are significantly reduced in the injection of A549 MALAT1 KO cells compared to A549 MALAT1 wild-type (WT) cells [50]. MALAT1 also directly binds with miR-200a to promote cell proliferation and induce gefitinib resistance in A549 cells [61]. An additional study showed that MALAT1 could promote cisplatin (DDP) resistance by sequestering miR-101-3p and upregulating the expression of myeloid cell leukemia 1 (MCL1) in LC cells [62]. Overall, MALAT1 plays an important role in the progression of LC through multiple mechanisms, and it could serve as a potential biomarker and target for the diagnosis and treatment of LC. However, the role of MALAT1 in regulating cancer stemness and post-translational modifications needs to be further studied. In particular, the mechanisms for MALAT1 activation in LC tissues still need further exploration.

### 3.2. H19

H19 is a 2.3 kb intergenic and maternally-expressed lncRNA located on chromosome 11p15.5, which is also one of the first identified imprinting lncRNA [40,41,46,63]. H19 includes five exons and four introns, and plays an important role in embryonic and tumor development [46,63,64]. H19 is reported to be associated with multiple cancers such as LC, GC, CRC, pancreatic cancer (PC), ovarian cancer (OC), neuroblastoma (NB), and bladder cancer [46,65].

Studies have revealed that the expression of H19 is significantly increased in LC tissues when compared with that in adjacent tissues [40,66]. H19 overexpression is associated with carcinogenesis from early stages to metastasis, reduced disease-free survival (DFS) time, and poor prognosis in LC [4,41,46,67]. Plasma level of H19 is also significantly increased in NSCLC patients, which could be a clinical serological biomarker for complementary diagnosis of NSCLC [68]. In the Chinese population, it has been reported that H19 rs2107425 is significantly related to LC susceptibility [69], and H19 SNP rs217727 is strongly associated with susceptibility to LSCC and LAD [69]. Kaplan–Meier analysis was used to evaluate that higher expressions of H19 and miR-21 are correlated with shorter OS in NSCLC patients, suggesting that H19 and miR-21 together may be involved in the development of LC [70].

H19 plays an important role in promoting cell proliferation and differentiation, cell migration and invasion, and EMT [40]. Zhou et al. reported that knockout of H19 obviously inhibits the proliferation of LC cells in vitro and in vivo [70]. Zhao et al. indicated that the overexpression of H19 sponged and inhibited miR-200a function to upregulate zinc finger E-box binding homeobox 1 (ZEB1) and zinc finger E-box binding homeobox 2 (ZEB2), thereby promoting proliferation, migration, invasion, and EMT in LC [40]. It has been demonstrated that upregulation of H19 can induce cell proliferation, invasion, and EMT occurrence by acting as an miRNA sponge and regulating miRNA-203-mediated EMT [66]. Additionally, H19 can induce cell proliferation by promoting the expression of pro-oncogene lin-28 homolog B (LIN28B) via the inhibition of miR-196b expression in A549 and H1299 LC cell lines [71]. H19 can also compete with miR-107 to bind neurofibromin 1 (NF1), which stimulates the development of NSCLC [42,72]. Additionally, H19/miR-29b-3p/signal transducer and activator of the transcription 3 (STAT3) signaling pathway is also considered to be involved in cell proliferation, viability, apoptosis, and EMT of LAD Calu-3 and NCI-H1975 cell lines [73]. H19 also promotes NSCLC metastasis by activating some cellular signaling pathway proteins, including metastasis associated in colon cancer 1 (MACC1), epidermal growth factor receptor (EGFR), β-catenin, and extracellular-signal-regulated kinase 1/2 (ERK1/2) [64]. Overexpression of H19 promotes cell proliferation, migration, and invasion by upregulating the expression of EMT markers, including snail family transcriptional repressor 1 (SNAI1), N-cadherin, and Vimentin [42,63,64,66,74]. Moreover, upregulation of H19 can induce A549 cells resistance to DDP [74,75]. SNP rs2107425 in H19 is also reported to be associated with resistance to platinum-based chemotherapy in SCLC [69]. Oncogene c-Myc has been shown to be associated with H19 regulation in NSCLC [76]. c-Myc induces H19 expression and then downregulates miR-107 to promote mitotic progression of the NSCLC cell line [76]. The in vivo and in vitro experiments revealed that c-Myc can bind to the promoter of H19 and then activate its transcription [77]. Collectively, activation of H19 promotes the progression of LC and it could also serve as a serological biomarker in diagnosing LC. Further studies might concentrate on the role in cancer stemness. Particularly, how to repress the expression of H19 in LC tissues represents a novel strategy for treating LC.

### 3.3. Taurine Upregulated Gene 1 (TUG1)

TUG1 is a 7.1 kb intergenic lncRNA located on chromosome 22q12 [78,79,80]. TUG1 was firstly discovered to play a crucial role in mouse retinal development [78,79,81]. After that, TUG1 was found to have an important role in other cancers such as LC, HCC, BC, OC, GC, CRC, esophageal squamous cell carcinoma (ESCC), osteosarcoma, glioma, and bladder cancer [78,82]. 

TUG1 is upregulated in the majority of the above cancer tissues but downregulated in LSCC and LAD tissues [78,79,80,82,83,84], suggesting that it may have tissue-specific expression patterns and a role in different human tumors [41]. NSCLC patients with high expression of TUG1 have a better prognosis [78]. In support, lower expression of TUG1 is associated with larger tumor size, advanced tumor lymph node metastasis (TNM) stage, and poorer OS in NSCLC patients [80]. Moreover, univariate and multivariate analysis revealed that TUG1 can serve as an independent predictor for OS in NSCLC patients [80]. However, TUG1 is found to be upregulated in SCLC tissues compare with that in adjacent normal tissues and its upregulation is associated with poor prognosis [85]. An additional study also revealed that TUG1 is upregulated in LAD cells and serum samples, and a high level of TUG1 is positively correlated with tumor size, TNM stage, lymph node metastases, and distant metastasis [86].

TUG1 expression is induced by the wild-type p53 in three NSCLC cell lines (A549, SK-MES-1, and NCI-H1299), and TUG1 knockdown increases cell proliferation in vitro and in vivo [80,84]. Additionally, TUG1 can bind with PRC2 in the promotor region of Elav-like family member 1 (CELF1) to inhibit the expression of the genes involved in cell cycle [87]. Moreover, TUG1/PRC2 complex could also bind to the homeobox B7 (HOXB7) promoter and activate its expression, thereby activating the Akt and mitogen-activated protein kinase (MAPK) pathways in NSCLC tissues [80]. However, TUG1 knockdown significantly inhibits cell proliferation, migration, and invasion and promotes cell apoptosis and cell cycle arrest in three SCLC cell lines (NCI-H69, NCI-H446, NCIH69AR) by regulating LIMK2b (a splice variant of LIM-kinase 2) expression via binding with enhancer of zeste 2 polycomb repressive complex 2 subunit (EZH2) [85]. TUG1 can also induce SCLC cell resistance to chemotherapeutic drugs, including DDP, adriamycin (ADM), and vepeside (VP-16) in vitro and in vivo [85]. In general, TUG1 plays an important role in the development of LC. However, further studies are needed to clarify the distinct mechanisms of TUG1 in NSCLC and SCLC.

### 3.4. Maternally Expressed Gene 3 (MEG3)

MEG3 is 6.9 kb in length and acts as a tumor suppressor gene located on chromosome 14q32.2 within the DLK1-MEG3 locus [41,88,89]. MEG3 is expressed in many normal tissues and is considered to be involved in many cellular processes [90]. Compared with adjacent non-tumor lung tissues, MEG3 is significantly downregulated in NSCLC tissues [90]. A meta-analysis declared that overexpression of MEG3 exhibited better prognosis in NSCLC patients, suggesting that MEG3 could serve as a prognostic factor of NSCLC patients [91]. Additionally, the MEG3 rs4081134 SNP is strongly associated with LC susceptibility in the Chinese population [88]. Moreover, MEG3 is also considered to be inhibited in other cancers, including HCC, NB, glioma, meningioma, bladder cancer, and hematological malignancies [4,88,92,93].

MEG3 is significantly downregulated in LC cell lines A549 and HCC823 [94]. Upregulation of MEG3 inhibits the viability, proliferation, and autophagy in H292 and A549 cells [90,95]. Upregulation of MEG3 increases cell apoptosis by suppressing the expression of apoptosis inhibitory protein B-cell lymphoma-2 (Bcl-2) and promoting apoptosis-promoting factor BCL2 associated X (Bax) in A549 cells [94]. MEG3 overexpression suppresses cell proliferation and induces apoptosis by activating the expression of p53 in vitro [89,90]. The retinoblastoma tumor suppressor (pRb) pathway, which is important in regulating cell cycle progression and cell proliferation, is revealed to inhibit cell proliferation by activating MEG3 expression in A549 and SK-MES-1 in LC cells [96]. In addition, the MEG3/microRNA-7-5p/BRCA1 regulatory network is also verified to be essential in NSCLC [94]. 

The drug resistance to chemotherapy drug vincristine is negatively associated with the expression of MEG3 in vitro [95]. Upregulation of MEG3 improves the sensitivity of DDP-resistant NSCLC cells to DDP treatment via inhibiting cell proliferation and inducing cell apoptosis modulated by the miR-21-5p/SRY-box transcription factor 7 (SOX7) axis [97]. MEG3 knockdown could also induce DDP resistance in A549/DDP cells by activating the Wnt/β-catenin signaling pathway [98]. Overall, these findings suggest that MEG3 could repress the development of LC, and it may also act as a therapeutic target for LC. Further study is also needed to clarify the mechanism(s) of MEG3 repression in LC tissues.

### 3.5. Actin Filament Associated Protein 1 Antisense RNA 1 (AFAP1-AS1)

AFAP1-AS1 is a 6.8 kb antisense lncRNA, which is located on chromosome 4p16.1. AFAP1-AS1 is transcribed from AFAP1 in the antisense direction and can also affect the expression of AFAP1 [43]. AFAP1-AS1 plays an important role in the progression of LC and other cancers, including LC, HCC, GC, PC, CRC, OC, and gallbladder cancer (GBC) [43]. 

AFAP1-AS1 expression is positively correlated with tumor pathological grade, TNM staging, smoking history, infiltration degree, distant metastasis, and clinical outcomes in NSCLC patients [91,99,100,101]. High expression of AFAP1-AS1 is correlated with advanced lymph node metastasis and reduced survival time [102]. A meta-analysis report showed that increased expression level of AFAP1-AS1 is a strong predictor of poor OS in NSCLC patients [91]. These findings suggest that AFAP1-AS1 could serve as a prognostic biomarker for NSCLC.

AFAP1-AS1 is upregulated in NSCLC tissues and cell lines [91,99,100,101,102,103,104,105,106,107]. Inhibition of AFAP-AS1 suppresses cell growth and migration and promotes apoptosis of NSCLC cells in vitro [103,104]. AFAP1-AS1 knockdown leads to a significant increase in keratin 1 (KRT1) expression in vitro, suggesting that the oncogenic activity of AFAP1-AS1 is mediated by suppressing KRT1 expression [104]. Moreover, AFAP1-AS1 knockdown also inhibits tumor growth of LC in BALB/c nude mice [104]. In contrast, overexpression of AFAP1-AS1 promotes proliferation, migration, and invasion, and inhibits apoptosis of NSCLC cell lines by upregulating the interferon regulatory factor (IRF) 7 and retinoid-inducible protein (RIG)-I-like receptor signaling pathways [105]. AFAP-AS1 can also promote NSCLC cell growth by recruiting EZH2 to epigenetically downregulate p21 expression [101]. In general, AFAP-AS1 promotes LC cell proliferation and migration, and inhibits cell apoptosis via multiple signaling pathways. 

## 4. LncRNAs in Breast Cancer (BC)

According to the Global Cancer Statistics 2018, the incidence of BC in both sexes is 11.6%, which is almost as much as that of LC [1]. It is estimated that BC is the second cause of cancer-related deaths, and has caused 626,679 deaths in 2018. BC incidence is also ranked the first among women, with an incidence of 24.2% among all female cancers [1]. Based on histological features, BC can be divided into three types: Hormone-receptor-positive, human epidermal growth factor receptor-2 overexpressing (HER2+), and triple-negative breast cancer (TNBC) [108]. Herceptin, tamoxifen, and trastuzumab have been commonly used for clinical treatment of hormone-receptor-positive BC and HER2-positive BC, and have achieved good clinical outcomes to a certain extent. However, due to chemoresistance of BC and limitations of the use of chemotherapy for TNBC, high mortality in BC still remains [108,109]. It has been reported that a large number of lncRNAs are involved in the pathogenesis of BC.

### 4.1. HOX Transcript Antisense Intergenic RNA (HOTAIR)

HOTAIR is located on human chromosome 12q13.13, and is a 2158 bp intergenic lncRNA situated inside the Homeobox C (HOXC) gene. HOTAIR is among the first lncRNAs reported to be involved in the development of cancer [110]. HOTAIR expression level is increased in multiple cancers, including BC [111], LC [112], GC [113], and NB [114]. Several studies have revealed that the high level of circulating HOTAIR might be a potential biomarker for BC [115,116,117]. Further study has suggested that the plasma HOTAIR level is positively associated with lymph node metastasis and estrogen receptor, and reduced after surgery in BC patients [118]. Genetic polymorphisms of HOTAIR are associated with BC risk in different races [119,120,121]. The latest findings suggest that HOTAIR-containing exosomes could be detected in BC patients’ blood [122,123]. Moreover, serum exosomal HOTAIR levels are higher in BC patients than in healthy subjects [122].

The expression of HOTAIR is important for BC cell growth, and loss of HOTAIR induces apoptosis in MCF-7 cells [124]. The promoter region of HOTAIR contains several estrogen response elements (EREs), and HOTAIR transcription is strongly induced by estrogen (E2) [124]. E2 could stimulate the expression of HOTAIR through G-protein-coupled estrogen receptor-1 (GPER) in TNBC cells [125]. However, another study found that high concentration of E2 could inhibit HOTAIR expression, and HOTAIR might act as a target gene of ER-mediated transcriptional repression. Moreover, HOTAIR is also involved in tamoxifen resistance of BC [126]. In ER-positive BC patients, HOTAIR expression is an independent biomarker for the prediction of metastasis risk [127]. Another study in ER-negative patients implied that HOTAIR could also be a marker for lymphatic metastases, rather than hematogenous metastases [128].

HOTAIR plays a critical role in regulating BC cell apoptosis, proliferation, migration, and invasion. Knockdown of HOTAIR suppresses cell proliferation and promotes apoptosis in several BC cell lines [129,130]. Moreover, downregulation of HOTAIR decreases the migration and invasion ability in MCF-7 cells [129]. MicroRNAs are the common target genes of HOTAIR. miR-20a-5p expression is negatively associated with HOTAIR in BC cells. Further study has revealed that HOTAIR targets miR-20a-5p to affect BC cell apoptosis, cell growth, and metastasis [130]. Bcl-2-like protein 2 (Bcl-w) belongs to the bcl-2 family, which plays a crucial role in cell migration and invasion, and cell survival in cancers [131,132]. In BC cells, HOTAIR can increase the expression of Bcl-w by sequestering miR-206 to promote cell proliferation [133]. A current study revealed that HOTAIR regulates one cell surface glycosaminoglycans-chondroitin sulfotransferase CHST15 to participate in BC cell invasion and migration [134]. Moreover, upregulation of HOTAIR also mediates the effects of transforming growth factor-β1 (TGF-β1) on the EMT process of BC cells [135].

In the MDA-MB-231 BC cell line, overexpression of HOTAIR promotes the expression of stemness-related genes such as SOX1, SOX10, aldehyde dehydrogenase 2 family member (ALDH2), and CD44 [135]. When compared with controls, knockdown of HOTAIR markedly inhibits colony formation and the self-renewal capacity of CSC-MCF-7 cell [136]. HOTAIR can decrease the expression of miR-7 in breast CSCs (BCSCs) and BC patients by inhibiting homeobox D10 (HoxD10) expression, thus accelerating the invasiveness and metastasis of BC [137]. Moreover, a study revealed that mixed lineage leukemia (MLL)-family of histone methylases MLL1 and MLL3, histone acetylases calcium-binding protein (CBP), and p300 can bind to the HOTAIR promoter ERE regions in the presence of bisphenol-A (BPA) and diethylstilbestrol (DES) in MCF-7 cells, thus mediating post-translational histone modifications, including histone H3K4-trimethylation and acetylations in the promoter histones, finally resulting in HOTAIR activation [138]. Overall, HOTAIR participates in cell proliferation, apoptosis, migration and invasion, EMT, and stemness through multiple pathways, and activation of HOTAIR can contribute to the development of BC. In particular, HOTAIR is important for maintaining the stemness phenotype of BC cells.

### 4.2. Growth Arrest-Specific 5 (GAS5)

GAS5 is localized on chromosome 1q25.1, and was originally isolated from the NIH 3T3 cell line [139]. Although GAS5 does not encode protein, multiple small nucleolar RNAs (snoRNAs) can be produced by GAS5 [140]. In a variety of tumors, including lung [141], liver [142], colorectal [143], and breast cancer [144,145,146], the expression of GAS5 is significantly decreased, suggesting that GAS5 might be a tumor-suppressing lncRNA. The expressions of GAS5 decreases in BC patient tissues, and the low GAS5 levels are associated with high histological grading and advanced TNM stage [144]. In addition, the expression levels of GAS5 in tumor tissues of young BC women (ages <45 years) are significantly lower than in that of old BC women patients (ages >45 years) [147]. Plasma levels of GAS5 in BC patients are negatively related with Ki67 proliferation index before surgery, indicating that plasma GAS5 may be a potential prognosis marker for BC patients [148]. GAS5 can also be detected in secreted exosomes, and the accumulation of GAS5 in exosomes may act as a biomarker in BC cells apoptosis [149]. A genetic variant of rs145204276 in the GAS5 promoter region is associated with a reduced risk of BC [150]. 

MiR-21 is an important oncogene in various malignant tumors [151,152,153]. GAS5 has a negative correlation with miR-21, suggesting that it may target miR-21 to inhibit cell growth and invasion in BC cells [154]. Overexpression of GAS5 inhibits proliferation and promotes apoptosis in TNBC cells. It has been further demonstrated that GAS5 may act as a ceRNA targeting miR-196a-5p to delay TNBC progression [155]. Moreover, GAS5 is a direct target gene of miR-221/222. In BC cells and xenografts mouse models, miR-221/222 inhibits the expression of GAS5 to promote cell proliferation, suppresses cell apoptosis, and increases tumor growth in vivo [145]. 

GAS5 is associated with drug and chemotherapy resistance in BC. Trastuzumab administration is a therapy method in clinic to treat HER2-positive, early-stage, and metastatic BC, but the effectiveness is limited [156]. The level of GAS5 in trastuzumab-resistant SKBR-3/Tr cells and trastuzumab-treated patients is significantly declined. GAS5 negatively regulates miR-21 to promote phosphatase and tensin homologs (PTEN) in the process of trastuzumab resistance of BC [144]. Decreased expression of GAS5 is also observed in tamoxifen-resistant MCF-7R cells. GAS5 overexpression enhances the sensitivity of BC cells to tamoxifen [157]. Another study revealed that a decrease of GAS5 expression weakens the effect of classical chemotherapy drugs on apoptosis of BC cells [158]. Overall, a decrease in GAS5 increases the proliferation and migration of BC cells, and induces resistance of BC cells to chemotherapy drugs. Activating GAS5 expression represents a novel strategy for treating BC.

### 4.3. H19

The expression of H19 is remarkably increased in BC tissues [159,160] and cells [160,161]. Additionally, the expression of H19 is upregulated in ER-positive BC tissues and cells than in ER-negative BC tissues and cells [162]. Plasma H19 level is positively correlated with lymph node metastasis, the expression of ER and progesterone receptor (PR), and significantly reduced after operation [159]. Genetic polymorphism rs217727 of H19 is associated with ER-positive and HER2-positive [163], HER2-negative, and hormone receptor-positive-HER2-negative [164] BC patients in the Chinese population. Moreover, rs2071095 of H19 is also correlated with BC risk in ER-positive patients in the Chinese population [165]. The SNPs of H19 are predicted to be risk factors of BC in the Iranian population [166,167].

Abolished H19 expression suppresses cell survival and blocks estrogen-induced cell growth in BC cells [162]. Knockdown of H19 also inhibits BC cells proliferation, invasion, and migration, and promotes cell cycle arrest and apoptosis [160]. In contrast, H19 overexpression promotes cell proliferation [161,162] and migration [161] in BC cell, and increases tumor growth and metastasis in vivo [161]. Mechanistically, H19 promotes BC cell proliferation and invasion through sponging miR-152, which targets DNA methyltransferase 1 (DNMT1) [168] and regulates miR-138 and SOX4 expressions [160]. H19 is also found to be involved in EMT and mesenchymal-epithelial transition (MET) by differentially sponging miR-200b/b and let-7b in BC cells [169]. In TNBC cells, H19 is regulated by lncRNA papillary thyroid carcinoma susceptibility candidate 3 (PTCSC3) to stimulate cell proliferation [170]. Vennin et al. reported that ubiquitin ligase E3 family, c-Cbl and Cbl-b, are direct targets of miR-675. H19/miR-675 can downregulate the expression of c-Cbl and Cbl-b, and increase the activation of tyrosine kinase receptors and downstream Akt and ERK pathways, thereby enhancing BC cell aggressiveness [161]. H19 is suggested to be involved in the maintenance of CSC characteristics in BC cells in vitro and in vivo, and H19/let-7/LIN28 double-negative feedback loop is considered to be involved in regulating BCSC maintenance [171].

H19 has been also reported to be involved in chemoresistance in BC. In paclitaxel-resistant TNBC cells, the expression of level H19 is significantly upregulated, and knockdown of lncRNA H19 enhances paclitaxel sensitivity and promotes apoptosis by suppressing the Akt signaling pathway in TNBC cells [172]. The expression of H19 is also elevated in ERα-positive BC cells, but not in ERα-negative BC cells. H19 can be regulated by ERα, and targets pro-apoptotic gene BIK to cause paclitaxel resistance of ERα-positive BC cells [173]. In addition, Cullin 4A (CUL4A) is an ubiquitin ligase component that is suggested to participate in multiple drug resistance in BC cells [174]. Zhu et al. found that the mechanism of doxorubicin resistance in BC cells is partly associated with the activation of the H19-CUL4A-ABCB1/MDR1 pathway [175]. H19 is also reported to be involved in endocrine therapy resistance (ETR) by activating the Notch and c-MET receptor signaling pathways [176]. Inhibition of H19 improves tamoxifen sensitivity by suppressing β-catenin in tamoxifen-resistant BC cells [177]. Overall, H19 plays multiple important roles in the development of BC, such as drug resistance, stemness regulation, and post-translational modifications. Further probing of the mechanism of H19 activation in BC tissues will shed light on the pathogenesis and treatment of BC.

### 4.4. Urothelial Carcinoma Associated 1 (UCA1)

UCA1 is a 2314 bp lncRNA composed of three exons and two introns located in human chromosome 19p13.12 [178,179]. UCA1 was firstly identified to be over-expressed in bladder cancer and was considered to function as a biomarker for the bladder cancer diagnosis [180]. Several studies have demonstrated that UCA1 expression is also upregulated in multiple types of other cancers, which include CRC, LC, BC, HCC, GC, and ESCC [178,181]. When compared with healthy tissues, the expression of UCA1 is significantly increased in BC tissues [182]. In metastatic BC patients, UCA1 is upregulated in tumor tissues, and the level of UCA1 is negatively correlated with prognosis [183]. UCA1 level is also positively associated with pathological grade and mortality of BC patients [184]. Plasma UCA1 level is significantly elevated in TNBC patients, indicating that it may act as a biomarker for TNBC-specific diagnosis [185]. 

P27 (Kip1) is a tumor suppresser, and its level is decreased in BC tissues [186]. UCA1 could interact with heterogeneous nuclear ribonucleoprotein I (hnRNP I) to repress p27 expression and promote cell growth and tumorigenesis in BC cells. In BC patients’ tissues, UCA1 level is negatively associated with p27 expression level [187]. UCA1 also directly interacts with miR-143 to inhibit its expression and downstream pathway to induce proliferation and inhibit apoptosis of BC cells [188]. Knockdown of UCA1 inhibits Akt phosphorylation and diminishes invasiveness of tumor cells induced by macrophage [189]. Additionally, the TGF-β/SLUG/E-cadherin pathway could play crucial roles in the progression of EMT in BC [190]. Li et al. discovered that UCA1 collaborates with lncRNA AC026904.1 to activate SLUG and then induces EMT and tumor metastasis [183]. UCA1 knockdown upregulates the expression of E-cadherin and decreases the expressions of β-catenin, cyclin D1, and MMP-7, indicating that UCA1 promotes BC cell EMT and invasion partly through β-catenin [191].

Tamoxifen is usually used to treat ER-positive BC patients, but the resistance of tamoxifen is still a question to be resolved [192]. When compared with tamoxifen-sensitive cells, the expression of UCA1 in tamoxifen-resistant BC cells is significantly increased [182,184,193], and inhibition of UCA1 improves tamoxifen sensitivity in BC cells [182,184]. Further studies have revealed that UCA1 might regulate multiple signaling pathways, including EZH2/p21, phosphatidylinositol 3′–kinase/protein kinase B (PI3K/Akt), mammalian target of rapamycin (mTOR), and Wnt/β-catenin, to induce tamoxifen resistance in BC cells [182,184,193]. UCA1 expression is also elevated in ER-positive BC cells, and a high level of UCA1 sponges miR-18a to attenuate tamoxifen sensitivity in BC cells [194]. Xu et al. found that UCA1 exists in exosomes released from tamoxifen-resistant LCC2 cells. Exosomes containing UCA1 isolated from LCC2 cells can promote tamoxifen resistance in ER-positive MCF-7 cells [195]. UCA1 is also involved in regulating trastuzumab sensitivity in BC cells by targeting the miR-18a/Yes-associated protein 1 (YAP1) axis [196]. Collectively, UCA1 promotes the migration and proliferation and involves in tamoxifen resistance in BC cells. Inhibiting UCA1 expression might be a potential strategy for treating BC.

### 4.5. LincRNA-ROR 

LincRNA-ROR is a 2.6 kb lncRNA, which is located at 18q21.31 [197,198]. The expression of LincRNA-ROR is significantly elevated in BC tissues and cell lines [199,200]. When compared with healthy controls, plasma lincRNA-ROR level is increased in BC patients, and its level is also correlated with ER [199] and lymph node metastasis [199,200]. Another study showed that the expression of lincRNA-ROR is higher in invasive breast tumors than that in early stage tumors [201]. Furthermore, circulating lincRNA-ROR level is decreased in BC patients after surgery [199].

In TNBC tissues [202,203] and cell lines [201,202,203], the level of lincRNA-ROR is increased. Overexpression of lincRNA-ROR in TNBC cell line MDA-MB-231 promotes cell invasion and metastasis, while inhibition of lincRNA-ROR expression exerts the opposite effects [202,204]. Additional studies have illustrated that lincRNA-ROR acts as a ceRNA sponging miR-145 to remove the restraining effect to small GTPase ADP-ribosylation factor 6 (ARF6) and Mucin1, and then stimulates TNBC cell invasion [201,202]. Hou et al. discovered that lincRNA-ROR also sponges miR-205 to prevent the degradation of EMT inducer ZEB2, causing BC cell growth [204]. Duru et al. pointed out that the SOX2/OCT4/lincRNA-ROR signaling axis is upregulated in aggressive clones from the CD49f^+^/CD44^+^/CD24^−^ MCF10DCIS stem cell population to regulate self-renewal of CSCs in BC [205].

Silencing of lincRNA-ROR inhibits BC cell proliferation, increases sensitivity to tamoxifen, and represses the activation of the PI3K/Akt/mTOR signaling pathway in MDA-MB-231 cell [203]. Inhibition of lincRNA-ROR also increases the sensitivity of BC cells to 5-Fluorouracil (5-FU) and paclitaxel by upregulating E-cadherin expression and downregulating Vimentin and N-cadherin expression. In contrast, overexpression of lincRNA-ROR induces resistance of BC cells to therapy drugs [200]. Overall, high expression of lincRNA-ROR promotes BC cell invasion and metastasis through targeting multiple pathways. Moreover, activation of lincRNA-ROR also causes resistance to anti-tumor drugs. To further illustrate the mechanism for lincRNA-ROR activation and to explore methodology for repressing it may hold promise for treating BC.

## 5. LncRNAs in Hepatocellular Carcinoma (HCC)

Liver cancer is the sixth most common malignancy worldwide, and the mortality rate is strikingly high. In 2018, there were approximately 782,000 deaths worldwide, which accounted for 8.2% of all cancer deaths [1]. The major primary liver cancer is HCC, which accounts for about 75–85% of total cases [1]. The main risk factors of HCC are mainly hepatitis B virus (HBV) or hepatitis C virus (HCV) infection, aflatoxin exposure, smoking, alcohol abuse, obesity, and type 2 diabetes [206]. Although several mechanisms of HCC have been proposed, it is still hard to diagnose HCC at an early stage. Recently, it had been reported that lncRNAs may be a new class of biomarker and a therapeutic target for HCC [207]. 

### 5.1. HOTAIR

HOTAIR has been considered to be a risk factor of HCC [112,208]. In HCC, the expression of HOTAIR is upregulated and its upregulation is associated with poor prognosis [112,209,210,211]. A current case control study showed that HOTAIR SNPs (rs12427129 and rs3816153) are the important predisposition factors for HCC in the south Chinese population [212]. 

Inhibition of HOTAIR represses the proliferation, migration, and invasion of HCC cells [112,210,211,213]. SiRNA suppression of HOTAIR reduces cell viability and invasion in HepG2 cells [112], and suppresses cell proliferation in Bel7402 cells with the decrease of matrix metalloproteinase-9 (MMP9) and vascular endothelial growth factor protein (VEGF) levels [210]. HOTAIR silencing also inhibits the activation of STAT3 and cyclin D1 expression to inhibit cell cycle [214]. HOTAIR can suppress RNA binding motif protein 38R (RBM38) to promote migration and invasion in HCC cells [213]. Overexpression of HOTAIR can elevate the expression of autophagy-related 3 (ATG3) and ATG7 expression to trigger autophagy and promote HCC cell proliferation [215], and induce EMT of liver cancer stem cells (LCSCs) by downregulating E-cadherin [216]. Li et al. reported that HOTAIR can promote human liver cancer stem cell (hLCSC) growth in vitro and in vivo, and plays an important role in hLCSC malignant growth by downregulating SET domain containing 2 (SETD2) [217]. HOTAIR also enhances the EMT of HCC cells by regulating ZEB1 via sponging miR-23b-3p [218]. PRC2 is an epigenetic repressor that can accelerate the development of HCC [219]. Moreover, HOTAIR directly interacts with the core subunit component of PRC2, recruits it to the promoter region of miR-218, and suppress its expression, then regulates chromatin remodeling and histone H3 lysine 27 (H3K27) trimethylation [220]. Knockdown of HOTAIR in HepG2 and Huh7 cells inhibits DNA methylation via the reduction of multiple DNA methyltransferases [221]. HOTAIR is also involved in the TGF-β1-induced multidrug resistance (MDR) process [222]. In HepG2 cells, HOTAIR can regulate Ras-related protein Rab-35(RAB35) expression and induce synaptosome associated protein 23 (SNAP23) protein phosphorylation to promote exosome secretion [223]. Collectively, a high level of HOTAIR is associated with poor HCC prognosis. Moreover, HOTAIR not only modulates cell proliferation, cell cycle, autophagy, and EMT, but also regulates hLCSC growth and drug resistance in the development of HCC. Further illustrating the mechanism of HOTAIR activation in tumor tissues holds promise for treating various cancers, including HCC.

### 5.2. Highly Upregulated in Liver Cancer (HULC)

HULC is located on chromosome 6p24.3, which is firstly characterized as a novel mRNA-like lncRNA, and HULC is shown to be significantly upregulated in human HCC [224]. A high level of HULC in human HCC tissues is associated with clinical stages, intrahepatic metastases, median survival, and recurrence ratio [225]. Another report indicated that the elevated expression of HULC in human HCC has a positive role in vascular invasion [226]. A variety of studies have suggested that HULC can be detected in blood samples of HCC patients [224,226,227,228] and its expression level is significantly higher in HCC patients than in healthy subjects [226,227,228]. Moreover, the expression of HULC is positively associated with OS and DFS [226]. These findings suggest that HULC may act as a novel biomarker for HCC diagnosis.

Knockdown of HULC in HCC cell lines represses cell proliferation, invasion, and migration, and promotes cell apoptosis [225], whereas HULC overexpression promotes Hep3B cells growth and increases the xenograft tumor weight and formation rate [229]. In HCC cancer cells, miR-203 [230] and miR-488 [231] target HULC to suppress cell proliferation and invasion. HULC also acts as an endogenous sponge to regulate miRNAs. HULC diminishes the expression and activity of miR-372 and sequesters miR-372 from its target gene protein kinase cAMP activated catalytic subunit beta (PRKACB) in liver cancer cells [232]. Transcription factor E2F transcription factor 1 (E2F1) can bind to the promoter of sphingosine kinase 1 (SPHK1). HULC not only increases the expression of E2F1, but also sequesters miR-107, which is an inhibitor of E2F1, to promote tumor angiogenesis [233]. HULC is also reported to accelerate EMT of HCC cells through the miR-200a-3p/ZEB1 signaling pathway [225]. HULC overexpression increases the expression of microtubule-associated protein 1 light chain II (LC3II) by targeting sirtuin-1 (Sirt1), and inhibits PTEN to activate the Akt/PI3K/mTOR pathway, thus accelerating HCC cells autophagy [229].

Hepatitis B Virus X protein (HBx) plays vital roles in the development of HCC. The expression levels of HULC are positively related with HBx in HCC patients, and HBx can induce the expression of HULC [234]. Another study revealed that HBx-induced HULC upregulation can be suppressed by metformin [235]. HULC also promotes hepatoma cell proliferation and growth by stimulating HBV replication [236]. In Chinese HBV persistent carriers, the SNP (rs7763881) of HULC is associated with the reduction of susceptibility to HCC [237]. Overall, HULC acts as a sponge to regulate different miRNA to promote HCC cell proliferation, migration, and invasion. Moreover, HULC is also involved in HBV-related HCC pathogenesis.

### 5.3. Nuclear Paraspeckle Assembly Transcript 1 (NEAT1)

NEAT1 is located on chromosome 11q13.1. NEAT1 has two transcription variants, which are termed as NEAT1_1 and NEAT1_2, respectively. Both of them are located in nuclear paraspeckles and constitute an important structure of paraspeckles [238]. When compared with the adjacent non-cancerous liver tissues, HCC tissues display upregulated NEAT1 expression [239]. High expression of NEAT1 is associated with tumorigenesis and metastasis of HCC [239]. In a Chinese population study, NEAT1 level was increased in HCC tissues, and served as an independent risk factor for prognosis of HCC [240]. Moreover, the serum level of NEAT1 is also significantly elevated in HCC patients [241]. 

In one study, it was reported that although cell growth is not affected after knockdown of NEAT1 in HepG2, cell invasion is significantly decreased [242]. Ling et al. further discovered that NEAT1 silencing inhibits HepG2 and HepB3 cell proliferation, migration, and invasion, and induces cell apoptosis [243]. NEAT1, together with miR-124-3p, could regulate adipose triglyceride lipase (ATGL) to promote HCC cell growth [244]. NEAT1 can also target miR-485 to regulate the STAT3 pathway in HCC [245]. miR-384 can directly bind with NEAT1, and overexpression of miR-384 suppresses NEAT1 functions [246]. NEAT1 might also be regulated by hypoxia-inducible factor-1α (HIF-1α) to further promote EMT and metastasis in HepG2 cells [247]. 

The balance between NEAT1_1 and NEAT1_2 may take part in the process of HCC cell malignant growth and metastasis [248]. Silencing of NEAT1_2 diminishes IL-6-induced STAT3 phosphorylation to repress HCC cell cycle progression, survival, and invasion [249]. Moreover, knockdown of NEAT1_2 may ameliorate HCC cells radiosensitivity [250]. NEAT1 is also involved in impairing anti-tumor drug (sorafenib) sensitivity. NEAT1 is suggested to negatively regulate miR-335 to induce drug resistance by activating the c-Met-Akt pathway [251]. The improved effect of immunotherapy may partly rely on the regulation of NEAT1 on CD8^+^T cell apoptosis and the cytolysis activity in HCC patients [252]. Collectively, a high level of NEAT1 promotes HCC tumorigenesis and metastasis, and induces resistance to anti-tumor drugs. The mechanism of post-translational regulation of NEAT1 needs further research.

### 5.4. Small Nucleolar RNA Host Gene 1 (SNHG1)

SNHG1 is located in the 11q12.3 region of the chromosome. Both public microarray data and a cohort study discovered that SNHG1 levels are significantly increased in HCC tissues, and the high expression of SNHG1 is positively associated with tumor size [253,254]. A meta-analysis confirmed that the SNHG1 level is significantly associated with reduced OS [255]. SNHG1 expression is only related with TNM stage and lymph node metastasis, but not tumor size, tumor subtype, or patient gender [255]. Circulating SNHG1 levels are remarkably increased in HCC patients [256], suggesting that it may serve as a biomarker for the diagnosis of HCC [254].

In HepG2 cells, the expression level of SNHG1 is remarkably increased, and the high level of SNHG1 downregulates miR-195 to promote HCC cell proliferation, invasion, and migration [257]. Another research showed that SNHG1 facilitates HCC cell proliferation by suppressing the expression of p53 and p53-related gene expression [253]. SNHG1 may also induce resistance to sorafenib. High expression of SNHG1 promotes sorafenib resistance by activating the solute carrier family 3 member 2 (SLC3A2)/Akt pathway in SR-HCC cells. In support, downregulation of SNHG1 increases sorafenib-induced cell apoptosis and autophagy by inhibiting the Akt pathway in SR-HCC cells. Sorafenib induces remarkable increase of miR-21 in nuclear fraction, suggesting that the role of SNHG1 in sorafenib resistance may be partly mediated by miR-21 [258]. Overall, high circulating levels of SNHG1 may act as a potential biomarker for early HCC diagnosis. SNHG1 could promote HCC cell proliferation and induce drug resistance via different signaling pathways. Further study is needed to determine the mechanisms of SNHG1 activation in HCC cells. 

### 5.5. LincRNA-p21

P21 is a canonical transcriptional target of p53 and plays a critical role in cell cycle and tumorigenensis [259]. lincRNA-p21 resides about 15 Kb upstream of the Cdkn1a (p21) gene on chromosome 17, and transcripts from the opposite orientation of the p21 gene [260]. Serum lincRNA-p21 level is negatively related with stages of liver fibrosis in HBV-infected patients [261], and low expression of lincRNA-p21 in human HCC tissues indicates advanced disease stage and poor survival [262]. Moreover, when compared with HCC patients, serum lincRNA-p21 level in liver metastatic cancer patients is significantly decreased [263]. 

In different tumor cell lines, lincRNA-p21 is induced by p53. lincRNA-p21 can interact with hnRNP-K to be involved in the apoptosis regulated by p53 [260]. Upregulation of lincRNA-p21 suppresses hepatic stellate cell proliferation and liver fibrogenesis [264], whereas knockdown of lincRNA-p21 in HCC cells exhibits an increase of proliferation ability. Further study suggests that lincRNA-p21 promotes endoplasmic reticulum stress to facilitate apoptosis of HCC cells [262]. One potential mechanism for lincRNA-p21-induced suppression of HCC cells migration and invasion is that lincRNA-p21 may negatively regulate the miR-9/E-cadherin pathway [265]. LincRNA-p21 may also mediate the Notch signaling pathway to participate in EMT [266]. Sorafenib could induce the expression of lincRNA-p21, both in vitro and in vivo, and lincRNA-p21 may also contribute to sorafenib-induced inhibition of HepG2 cell growth in vivo [262]. Collectively, lincRNA-p21 may act as a tumor-suppressing lncRNA, which inhibits HCC cell proliferation and induces apoptosis. Moreover, it is interesting to further determine whether or not lincRNA-p21 mediates the anti-tumor effects of p53.

## 6. LncRNAs in Colorectal Cancer (CRC)

CRC is one of the most common gastrointestinal tumors, with the second highest incidence among females (9.5%) and the third among males (10.9%) [1,267]. About 1,096,601 new cases were diagnosed and 551,269 people died from CRC worldwide in the year of 2018 [1]. In some developed counties, most CRC cases are screened at the early stage, and early endoscopic resection is used to effectively reduce the incidence of CRC. However, nearly half of new CRC cases are still diagnosed at the late stage [268], accompanied by a five-year survival rate of CRC with distant stage is only 11% [268]. In addition, low screening rate, lack of effective treatment, and chemotherapy drug resistance still exist, resulting in a high mortality rate and a low OS rate in CRC [268,269,270,271,272]. Although the specific pathogenesis of CRC remains unclear, the important roles of lncRNAs in CRC have been proposed.

### 6.1. Colon Cancer Associated Transcript 1 (CCAT1) 

CCAT1 is a newly discovered oncogenic lncRNA, 2628 bp in length, which is located on chromosome 8q24.21 [46,273]. CCAT1 is firstly found to be upregulated in colon cancer [274] and its expression is suggested to be epigenetically activated by c-Myc [46,275]. There are two isoforms of CCAT1: CCAT1-S and CCAT1-L. CCAT1-S is expressed in the cytoplasm, while CCAT1-L is exclusively expressed in the nucleus. CCAT1-S can be generated from CCAT1-L and there is a positive regulatory relationship between them [275]. CCAT1 is reported to be upregulated in many types of cancer, including CRC, LC, GC, HCC, ESCC, and acute myeloid leukemia (AML) [275,276]. Studies have shown that the expression of CCAT1 in CRC tissue and CRC cell lines is significantly increased than that in normal para-carcinoma or adjacent mucosa tissue and normal colon-derived cells [277,278,279,280,281]. A research reported that CCAT1 expression in more than 90% colon cancer tissue samples is at least five-folds higher than that in normal colon tissue [277]. Another study suggests that CCAT1 expression is not only upregulated in the early stage of adenocarcinoma of the colon, but also in the later stage of CRC, such as liver metastasis [274]. Additionally, CCAT1 is also increased in the blood of CRC patients when compared to healthy people [274]. A combination of high expressions of CCAT1 and HOTAIR in plasma of CRC patients has 85% in specificity for detecting CRC at the early stage [282]. Moreover, CCAT1 SNP rs67085638 C > T is associated with increased risk of CRC [283]. A high level of CCAT1 in CRC tissues is associated with poor relapse free survival (RFS) and OS, indicating that CCAT1 could serve as a potential prognostic biomarker in CRC [279].

Mechanistically, CCAT1 expression is considered to be associated with cell cycle, proliferation, apoptosis, migration, invasion, and EMT process in multiple cells [275]. Overexpression of CCAT1 promotes cell proliferation and invasion [284], while downregulation of CCAT1 inhibits cell proliferation, migration, and invasion, and results in cell cycle arrest in CRC cell lines [278,284,285]. CCAT1 also plays an important role in the EMT of CRC, with decreased expression of E-cadherin and increased expression of N-cadherin [278]. CCAT1 promotes c-Myc expression by inhibition of the MAPK pathway [286], and c-Myc can also promote CCAT1 transcription via binding to its promoter region [284]. Moreover, CCAT1 also acts as an enhancer-templated RNA to promote bromodomain and extraterminal (BET)-mediated c-Myc transcription in CRC [287]. Overall, a high level of CCAT1 is strongly associated with CRC progression. CCAT1 could be a potential biomarker for the diagnosis of CRC. Interrupting the cross regulation loop among CCAT1 and c-Myc might be the potential strategy for treating CRC. 

### 6.2. Colon Cancer Associated Transcript 2 (CCAT2)

CCAT2 is a 1752 bp lncRNA located on chromosomal region 8q24.21 [288,289,290]. CCAT2 is originally considered as an oncogenic lncRNA in CRC, which is significantly upregulated in CRC tissues and cell lines [279,291,292,293]. In addition, the expression level of CCAT2 in microsatellite-stable (MSS) CRC tissue samples is 10-folds higher than that in microsatellite-unstable (MSI-H) tumors and adjacent normal colon mucosa [291]. CCAT2 has also been demonstrated to be associated with other cancers, including GC, BC, NSCLC, and ESCC [288]. A study clarified that a high level of CCAT2 expression in CRC is associated with poor cell differentiation, deep tumor infiltration, increased lymph node metastasis, advanced TNM stage, and short DFS and OS [279,292]. The serum level of CCAT2 expression in CRC patients is suggested to be significantly increased compared with that in controls [294]. Moreover, CRC patients with CAAT2 rs6983267 GG genotype have higher serum CCAT2 levels than those patients with GT/TT genotypes, suggesting that CCAT2 is a novel potential non-invasive diagnostic and prognostic biomarker for CRC [295].

Overexpression of CCAT2 in CRC cells could promote tumor growth, migration, and invasion in vitro and vivo and also induce chromosomal instability in vitro [291]. There is a physical interaction between CCAT2 and transcription factor 7 like 2 (TCF7L2), resulting in increased transcriptional activity of MYC proto-oncogene and then enhancing the activity of the Wnt signaling pathway [291]. CCAT2 also suppresses miR-145 maturation by inhibiting pre-miR-145 export to the cytoplasm and then regulating colon CSCs proliferation and differentiation [293]. In general, high expression of CCAT2 is associated with tumorigenesis and it could be used as a novel biomarker for evaluating the prognosis in CRC patients. It is important to further determine the distinct mechanisms of CCAT1 and CCAT2 in the pathogenesis of CRC. Moreover, because CCAT1 and CCAT2 are located at the same region of the chromosome, to determine whether they share similar activation mechanism is also of significance.

### 6.3. Colorectal Neoplasia Differentially Expressed (CRNDE)

CRNDE is located on chromosome 16, and contains six exons: E1–E6 [296]. CRNDE is found to be overexpressed in colorectal adenomas and adenocarcinomas [297], and several other cancers, such as NSCLC, HCC, GC, OC, renal cell carcinoma (RCC), and glioma. [296]. A meta-analysis reported that overexpression of CRNDE is significantly associated with advanced TNM stage, increased lymph node metastasis, and poorer OS in patients with solid tumors, including CRC, suggesting that CRNDE could be used as a reliable prognostic factor in human cancer [298].

CRNDE is significantly upregulated in both CRC tissues and cell lines compared with that in adjacent tissues and normal colon cell line [299]. CRNDE knockdown inhibits the proliferation, migration, and invasion, and promotes apoptosis of CRC cells in vitro and in vivo [299,300,301,302]. CRNDE can promote CRC development by epigenetically suppressing the expressions of dual-specificity phosphatase 5 (DUSP5) and cyclin-dependent kinase inhibitor 1A (CDKN1A) [301]. Cytoplasmic heterogeneous nuclear ribonucleoprotein U-like 2 protein (hnRNPUL2) interacts and stabilizes CRNDE, and then promotes CRC cell proliferation and migration by activating its resistance to the audiogenic seizures (Ras)/MAPK signaling pathways [302]. The Wnt/β-catenin signaling pathway is reported to be crucial in CRC development, and its abnormal activation is discovered in more than 90% of CRC cases [303]. A study showed that CRNDE knockdown can suppress cell proliferation, reduce chemoresistance, and inhibit the Wnt/β-catenin signaling pathway by modulating the expression of miR-181a-5p [304]. Moreover, CRNDE knockdown can increase the sensitivity of CRC cells to oxaliplatin (OXA) by modulating E2F1 expression [299]. Overall, high expression of CRNDE is associated with the development of CRC, and CRNDE could be used as a potential prognostic factor in CRC. 

### 6.4. UCA1

A research clarified that the level of UCA1 in CRC tissues is positively correlated with CRC tumor size and advanced tumor stages [305]. A meta-analysis showed that high expression of UCA1 is significantly associated with poorer OS and DFS in patients with digestive system cancers, including CRC, suggesting that UCA1 could serve as an prognostic factor for predicting clinical outcome for patients with digestive system cancers [179]. Another meta-analysis explored that UCA1 might be a risk factor in predicting short OS and progression-free survival (PFS) in cancers, including CRC, NSCLC, GC, and OC [306]. Additional study suggests that overexpression of UCA1 is significantly related to CRC patients with lymph node metastasis [307]. The combination of HOTTIP, PVT1, and UCA1 is identified to be a clinical predictive panel to differentiate CRC patients with lymph node metastasis from CRC patients with non-metastatic lymph nodes and healthy people [308]. Moreover, exosomal UCA1 is detectable and stable in the serum of CRC patients, and circulating UCA1-containing exosomes could be helpful in predicting the clinical outcome of cetuximab-treated CRC patients. This suggests that UCA1 could be a potential biomarker for predicting cetuximab resistance in CRC patients [309].

UCA1 is demonstrated to be involved in the process of CRC tumorigenesis and progression, including cell proliferation, cell cycle, autophagy, apoptosis, and drug resistance [181,310]. UCA1 expression is higher in cetuximab-resistant CRC cells and their exosomes [309]. Knockdown of UCA1 inhibits proliferation and autophagy and enhances apoptosis in CRC cells [310]. The UCA1/mTOR axis is suggested to be induced by cancer-associated fibroblasts (CAFs) [311]. One study showed that homeobox B3 (HOXB3), a target gene of miR-28-5p, can mediate the effects of UCA1 in cell proliferation and invasion of CRC cells [305]. UCA1 can also induce 5-FU resistance by inhibiting miR-204-5p in CRC cells [181]. In summary, UCA1 activation causes CRC development, and induces drug resistance in CRC cells. However, further studies are still needed to clarify the underlying mechanisms.

## 7. LncRNAs in Hematological Malignancies

Hematological malignancies are one of the most common types of malignant tumors that are genetically divided into acute leukemia (AL), myelodysplastic syndromes (MDS), myeloproliferative neoplasms (MPNs), chronic lymphocytic leukemia (CLL), and lymphoma [312]. According to statistics, non-Hodgkin lymphoma accounted for the first place in the incidence of hematological malignancies, with a 2.8% incidence and a 2.6% mortality rate in 2018 [1]. Leukemia accounted for the first place in the mortality of hematological malignancies, with a 3.2% mortality rate accompanied by a 2.4% incidence rate in 2018 [1]. There is increasing evidence that lncRNAs also play an important role in both normal and malignant hematopoiesis, which involves blood cell differentiation, proliferation, and survival [313,314]. 

### 7.1. Plasmacytoma Variant Translocation 1 (PVT1)

PVT1 is an oncogenic lncRNA located at chromosome 8q24.2 downstream of c-Myc, and this region is thought to be cancer-related because of its function with c-Myc and rearrangement [315,316,317,318]. Studies have shown that PVT1 rearrangement can occur in both multiple myeloma (MM) and B-cell lymphomas (BCLs) with 8q24 abnormalities, accompanied by the incidence of rearrangement of 58.3% in MM patients and 37.5% in BCLs patients [317,319]. Overexpression of copy number of PVT1 has been reported to be involved in multiple types of human cancers, including GC, CRC, BC, OC, bladder cancer, and osteosarcoma [320,321,322,323,324]. It has been indicated that PVT1 expression in human GC and NSCLC tissues, and serum PVT1 levels in cervical cancer patients are increased, which are significantly correlated with TNM stage, lymph node metastasis, and poor prognosis, and might serve as a biomarker for diagnosis and prognosis of these cancers [325,326,327]. Recently, PVT1 oncogene has been explored to be crucial in the pathogenesis of hematological malignancies. 

Zeng et al. reported that PVT1 level in peripheral blood samples from patients with acute promyelocytic leukemia (APL) is significantly upregulated when compared with healthy donors, which indicates that PVT1 may act as a novel biomarker for APL diagnosis [328]. Izadifard et al. found that acute myeloid leukemia-M3 (AML-M3) patients have higher PVT1 expression than normal controls, and high-risk AML-M3 patients have higher expression levels of PVT1 than low- and intermediate-risk patients [315]. Another study suggested that the expression of PVT1 is upregulated in MM samples and cell lines when compared to control groups [329]. Yang et al. found that PVT1 can promote MM cell proliferation by reducing MM cell apoptosis through suppressing miR-203a expression [329]. In KG1 cell line (human acute erythroleukemia, AML-M6), apoptosis and necrosis are significantly induced by PVT1 inhibition [330]. Houshmand et al. have shown that compared with normal lymphocytes, the expression of PVT1 is significantly upregulated in hematologic malignancy cell lines. Knockdown of PVT1 can promote c-Myc downregulation, suppress cell proliferation, induce cell apoptosis and G0/G1 arrest in cell cycle in vitro. The authors further investigated that the expression of human telomerase reverse transcriptase (hTERT) protein is decreased after PVT1 inhibition in K562, TF-1, and HL-60 cell lines, which indicates that PVT1 can regulate the telomerase activity thereby influencing the ability of hematologic malignancy cells to divide infinitely [316]. These findings suggest that PVT1 is a potential therapeutic target for treating MM, AML, chronic myeloid leukemia (CML), and acute lymphoblastic leukemia (ALL) [316,331]. Overall, PVT1 participates in the pathogenesis of several types of hematological malignancies through regulating multiple processes, such as cell cycle, cell proliferation, and cell apoptosis. It is necessary to determine the mechanisms of PVT1 activation in hematological malignancies.

### 7.2. Deleted in Lymphocytic Leukemia 1 (DLEU1) and Deleted in Lymphocytic Leukemia 2 (DLEU2)

LncRNA DLEU1 and DLEU2 are located in the region of chromosome 13q14.3 [332,333], and they are reported to be involved in the pathogenesis of several types of solid tumors such as HCC [334], CRC [335], LSCC [336], PC [337], OC [338], etc. Nava-Rodríguez et al. reported that a CLL patient has genomic instability and a 13q14 deletion involving the DLEU1, DLEU2, and RB1 genes. This suggests that the combined detection of RB1 and DLEU gene is involved in the progression of CLL [339]. Additionally, DLEU2 is downregulated in patients with MM carrying del13 [340].

Lee et al. found that knockdown of DLEU1 in the human Burkitt lymphoma (BL) cell line results in a significant increase in cell proliferation and a reduction in apoptosis when compared to WT cells. Conversely, overexpression of DLEU1 in Raji cells causes a significant decrease in cell proliferation and an increase in apoptosis compared to control cells. The authors further showed that DLEU1 knockdown leads to a significant upregulation of anti-apoptotic genes and a downregulation of pro-apoptotic genes [332]. Additionally, they observed a chemoimmunotherapy resistant to rituximab (RTX) or cyclophosphamide (CTX) in DLEU1-KD BL cell compared with WT cells [332]. 

Researches have shown that DLEU1, DLEU2, and embedded miRNA cluster miR-15a/16-1 are frequently deleted in CLL, MM, mantle cell lymphoma (MCL), and some types of solid tumors [341,342,343,344]. DLEU1 and DLEU2 control the expression of epigenetic tumor suppressors located at chromosome 13q14.3, including the miR-15a/miR-16 family (miR-15a, miR-15b, miR-16, miR195, miR424, miR497), which can then modulate NF-κB activity [344]. One report demonstrated that DLEU2 overexpression inhibits cell proliferation and ability of colony forming in CLL cell through functional loss of miR-15a/miR-16-1 in vitro [341]. In support, deletion of miR-15a/16-1 promotes the proliferation of both human and mouse B cells by accelerating the G0/G1-S phase transition [343]. Furthermore, another research revealed that DLEU2 and miR-15a/16-1 cluster are regulated independently in pediatric AML, and the expression of miR-15a/16-1 cluster is not correlated with DLEU2 promoter DNA methylation [342]. Collectively, the presence of 13q14.3 deletion could be used for screening some types of hematological malignancies. In particular, DLEU1 and DLEU2 could serve as potential therapeutic target for hematological malignancies.

### 7.3. MEG3

MEG3 is significantly downregulated in AML patients’ tissue samples and cell lines when compared to normal controls, particularly in the Wilms’ tumor 1 (WT1) or ten–eleven translocation 2 (TET2) mutant AML patients [92,345,346]. DNA hypermethylation of CpG sites within the MEG3 promotor region can be seen in bone marrow (BM) samples from AML patients, and decreased TET2 expression level in AML patients may result in elevated MEG3 promoter methylation, which ultimately represses MEG3 expression and causes AML development [346,347]. Additionally, overexpression of MEG3 in MOLM-13 cell line inhibits cell proliferation and induces apoptosis and G0/G1 cell cycle arrest when compared with controls [345]. Moreover, MEG3 suppresses tumor growth through both p53-dependent and -independent pathways [345]. Furthermore, Li et al. reported that patients with different phases of CML had lower expressions of MEG3 and miR-147 when compared to the controls [348]. They further explored that overexpression of MEG3 and its target miR-147 could inhibit cell proliferation and induce apoptosis via regulating the Janus kinase (JAK)/STAT pathway in vitro [348]. Overall, MEG3 could serve as a novel therapeutic target for several types of hematological malignancies. 

### 7.4. MALAT1

Huang et al. reported that the expression of MALAT1 in patients with AML-M5 is significantly increased when compared with healthy controls. The OS of AML-M5 patients with high MALAT1 expression is significantly decreased compared with that of AML-M5 patients with low MALAT1 expression [47]. Upregulation of MALAT1 in MM patients is significantly associated with short PFS and OS [349,350]. MALAT1 expression is also obviously upregulated in CLL patients when compared to healthy controls [351]. 

The expression of MALAT-1 in patients with MM and MM cell lines is significantly higher than that in normal controls [349,350,352,353,354]. MiR-125b can regulate MALAT1 expression via the Notch1 pathway to modulate MM cell proliferation [355]. MALAT-1 knockdown significantly inhibits MM cell growth, and induces apoptosis and cell cycle arrest in vitro and in vivo [352,353]. Li et al. reported that compared with the blank group and the siNC group, the siRNA-MALAT-1 group had significantly decreased tumor volume and weight in tumor xenograft models in nude mice [356]. Hu et al. found that inhibition of MALAT1 also induced DNA damage and apoptosis in MM through binding to poly (ADP-ribose) polymerase 1/DNA ligase 3 (PARP1/LIG3) protein complexes, thereby inhibiting MM cell growth [354]. Conversely, overexpression of MALAT1 in MM markedly increases cell proliferation and represses apoptosis in vitro and in vivo [354]. MALAT-1 is also reported to promote autophagy in MM through upregulation of high mobility group box 1 (HMGB1) in MM [352]. Additionally, MALAT1 can positively modulate Forkhead box protein P1 (FOXP1) expression through sponging miR-509-5p in MM, thereby regulating MM cell growth [353]. 

The expression of MALAT-1 is significantly higher in diffuse large B cell lymphoma (DLBCL) cell lines than that in normal controls [356]. MALAT1 expression level is also markedly elevated in MCL cell lines and human MCL tumors when compared to normal controls, and MALAT1 level is negatively correlated with the prognosis and OS of MCL patients. Knockdown of MALAT1 can inhibit cell proliferation and induce cell cycle arrest and apoptosis in MCL cell lines [357]. Collectively, MALAT1 plays an important role in the pathogenesis of hematological malignancies and serves as a potential target for hematological malignancy diagnosis, prognosis, and treatment.

### 7.5. HOTAIR

The expression of HOTAIR is significantly upregulated in DLBCL tissues and cell lines when compared with normal controls. DLBCL patients with higher expression of HOTAIR have poorer OS [358]. Knockdown of HOTAIR inhibits cell proliferation and induces cell apoptosis and G2/M cell cycle arrest in DLBCL cells, possibly via suppressing the PI3K/Akt/mTOR nuclear factor-kappa B (NF-κB) pathway activation [358]. 

HOTAIR expression level is tested to be significantly upregulated in BM/peripheral white blood cell (PB) samples from AML patients when compared with AML-complete remission (CR) patients and healthy controls. Its level is markedly downregulated in AML patients after treatment [359,360,361]. High expression of HOTAIR is correlated with low OS and RFS of AML patients [360]. HOTAIR is found to be upregulated in AML cells [361], and knockdown of HOTAIR inhibits proliferation and induces apoptosis of AML cells in vitro and in vivo [359,361]. In acute leukemia (AL) patients, the expression of HOTAIR and its multiple downstream genes, including EZH2, lysine demethylase 1 (LSD1), DNA methyltransferase 3 alpha (DNMT3A), and DNMT3B, are also significantly increased [362]. 

HOTAIR can interact with miR-143, regulating CML cell proliferation, knockdown of HOTAIR, or overexpression of miR-143, inhibiting the phosphorylation of the PI3K/AKT/mTOR signaling pathway [363]. Moreover, research of HOTAR on CML resistance to MDR has also been reported. HOTAIR is markedly upregulated in BM samples of CML patients compared with health controls, and the multidrug resistance protein 1 (MRP1)-high expression group has higher expression of HOTAIR when compared with MRP1-low expression group [364]. Knockdown of HOTAIR increases the sensitivity of K562-imatinib-resistant cells to the imatinib treatment [364]. In summary, high HOTAIR expression is closely related with a poor prognosis in patients with leukemia and lymphoma. HOTAIR could serve as a potential biomarker for diagnosis, prognosis, and treatment of leukemia and lymphoma. So far, the roles of HOTAIR in cancer stemness, post-translational modifications, and drug resistance need to be further studied.

## 8. LncRNAs in Neuroblastoma (NB)

NB is the most common pediatric extracranial solid tumor, which derives from primitive sympathetic neural precursors, and accounts for more than 7–10% of all pediatric cancers and 15% of all pediatric cancer deaths [365,366,367,368]. Similar to the other cancers discussed above, some lncRNAs are reported to be involved in the pathogenesis of NB. Multiple analyses have revealed that SNPs of lnc-LAMC2 [369], LINC00673 [370,371], HOTAIR [114], and H19 [65] are significantly associated with increased susceptibility to NB in Chinese children.

### 8.1. MALAT1

Under hypoxic conditions, the expression of MALAT1 in NB cells promotes endothelial cell migration, invasion, and vasculature formation [372]. MALAT1 can upregulate AXL receptor tyrosine kinase expression in NB cells, thereby promoting cell migration and invasion [48]. Chen et al. reported that MALAT1 can promote neurite outgrowth in N2A cells via activating the MAPK/ERK signaling pathway [373]. Histone demethylase JMJD1A induces N-Myc-amplified NB cell migration and invasion by activating MALAT1 gene transcription [374]. 

### 8.2. SNHG1, SNHG7, SNHG16

The expression of SNHG7 is markedly higher in NB tissues than that in non-tumor tissues [375]. Kaplan–Meier survival analysis revealed that NB patients with high expression of SNHG1 [376], SNHG7 [375], and SNHG16 [377] have poor clinical outcomes, suggesting that these lncRNAs could act as prognostic biomarkers for predicting OS and event-free survival (EFS) in NB patients. Mechanistically, SNHG1 promotes α-synuclein aggregation, resulting in dopaminergic neuronal toxicity by modulating the miR-15b-5p/siah E3 ubiquitin protein ligase 1 (SIAH1) axis in human SH-SY5Y cells [378]. SNHG7 can enhance NB progression by regulating miR-653-5p/signal transducer and activator of transcription 2 (STAT2) feedback loop [375]. Knockdown of SNHG7 and SNHG16 can suppress cell proliferation, migration, invasionm and EMT, and induce cell cycle arrest at the G0/G1 phase in NB cells [375,377]. 

### 8.3. CASC15 and NBAT1

Cancer susceptibility 15 (CASC15) and neuroblastoma associated transcript 1 (NBAT1) are tumor suppressors that are located at the NB risk-associated 6p22.3 locus [379,380]. CASC15-003 and CASC15-004 are isoforms encoded by CASC15 [379]. Kaplan–Meier analysis indicated that patients with NB have lower expressions of CASC15 [379], NBAT1 [381], short CASC15 isoform (CASC15-S) [380], CASC15-003, and CASC15-004 [379] and exhibit poorer OS and EFS when compared with those with higher expressions of these lncRNAs. Researches have demonstrated that knockdown of CASC15 and NBAT1 in NB cell line can prevent neuronal differentiation through increasing nucleolar localization of ubiquitin specific peptidase 36 (USP36), activating SOX9 by regulating chromodomain helicase DNA binding protein 7 (CHD7) ubiquitination [379], and upregulating RE1 silencing transcription factor (REST/NRSF) [381]. Additionally, knockdown of NBAT-1, CASC15-003, and CASC15-S can induce NB cell proliferation, migration, and invasion in vitro and in vivo [379,380,381]. 

### 8.4. Pax6 Upstream Antisense RNA (Paupar)

LncRNA Paupar is transcribed from 8.5 kb upstream of the paired box 6 (Pax6). Reports have shown that Paupar knockdown can regulate the expression of multiple genes related to cell cycle, DNA replication, and mitosis, induce elevation of cell count in S and G2 phases, and increase neurite outgrowth in N2A cells [382]. Paupar can interact with chromatin regulatory protein kinesin-ii-associated protein (KAP1) that regulates a set of target genes involved in neural cell differentiation, nervous system development, cell-cell signaling, the RTK signaling pathway, and synaptic transmission in N2A cells [383]. Paupar can also participate in the transcriptional regulation of neuro-developmental genes [382].

### 8.5. Others

LINC01105 and hyperpolarization activated cyclic nucleotide gated potassium channel 3 (HCN3) are significantly upregulated, whereas MEG3 is downregulated in NB tissues when compared to control tissues [93]. It has been further revealed that they can modulate NB cell proliferation and apoptosis through HIF-1α and p53 pathways [93]. 

Kaplan–Meier curves analyses have shown that NB patients with high expression of Ets-1 promoter–associated noncoding RNA (pancEts-1) and hnRNPK are associated with poor OS and EFS. Significant higher pancEts-1 and hnRNPK transcription levels can be observed in primary NB tumors than that in normal dorsal ganglia tissues, and upregulation of pancEts-1 and hnRNPK is also correlated with NB cases with poor differentiation and advanced International Neuroblastoma Staging System (INSS) stage. Mechanistically, pancEts-1 can promote NB cell growth, invasion, and metastasis in vitro and in vivo through β-catenin transactivation and other mechanisms [384].

Zhang et al. found that X inactive specific transcript (XIST) is markedly upregulated in NB tissues when compared with that in control tissues. XIST can stimulate NB cell proliferation, migration, and invasion by epigenetically downregulating dickkopf WNT signaling pathway inhibitor 1 (DKK1) expression through induction of H3 histone methylation via EZH2 [385].

Knockdown of lncRNA myocardial infarction associated transcript (MIAT) reduces cell growth and migration, and enhances apoptosis in NB cell line when compared with that in control [386]. Atmadibrata et al. reported that linc00467 can increase cell viability and reduce apoptosis and cell cycle arrest in NB cells via suppressing DKK1 expression [387]. Moreover, nuclear forkhead box D3 antisense RNA 1 (FOXD3-AS1) suppresses NB cell growth, invasion, and metastasis by interacting with PARP1 protein to repress CCCTC-binding factor (CTCF) activation in vitro and in vivo [388].

Overall, multiple lncRNAs have been shown to be involved in the pathogenesis of NB. Further study is needed to explore the distinct roles and mechanisms of lncRNAs in the development of NB. Moreover, whether lncRNAs could serve as plasma biomarkers for predicting the risk of NB also awaits exploration. 

## 9. Conclusions and Perspective

LncRNAs play an important role in the pathogenesis of various malignant cancers by promoting cell proliferation, migration, and invasion at transcriptional, translational, and post-translational levels. In particular, several lncRNAs play important roles in the pathogenesis of various cancers, suggesting that they represent a new class of targets for treating different cancers (Figure 1 and Figure 2, Table 2). Additionally, circulating lncRNAs could serve as novel biomarkers for early diagnosis of various cancers. Further study of the roles and mechanisms of the reported lncRNAs in the development of cancer cell proliferation and invasion, in particular the mechanisms of lncRNA-induced resistance of cancer cells to chemotherapy drugs, will shed light on the treatment of various cancers. Furthermore, given the vast number of lncRNAs in humans, and the fact that only a few numbers of lncRNAs have function annotation, it is urgent to develop effective methods for identifying new important lncRNAs and to study their roles and mechanisms in the pathogenesis of cancers. 

MALAT1 plays an important role in the development of various cancers by modulating cell proliferation, migration and apoptosis. The symbol “↑” means activation, and symbol “⊥” represents repression.

HOTAIR plays an important role in the pathogenesis of various cancers by multiple signaling pathways. Targeting HOTAIR represents a novel strategy for treating various cancers. The symbol “↑” means activation, and symbol “⊥” represents repression.

## Figures and Tables

**Figure 1 cells-08-01015-f001:**
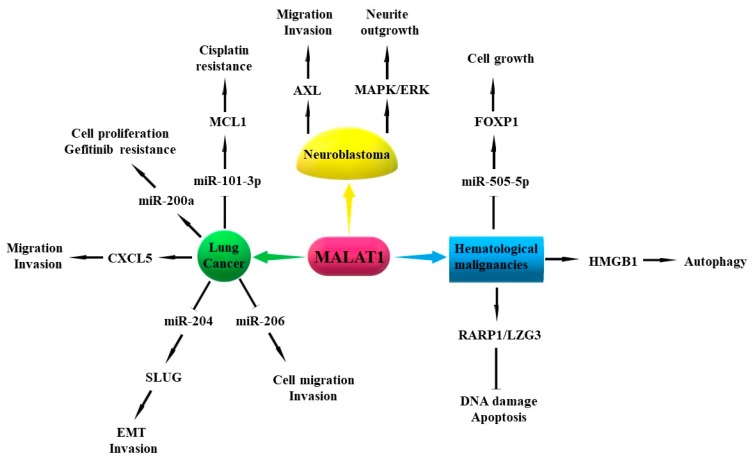
Proposed mechanisms of MALAT1 in various cancers.

**Figure 2 cells-08-01015-f002:**
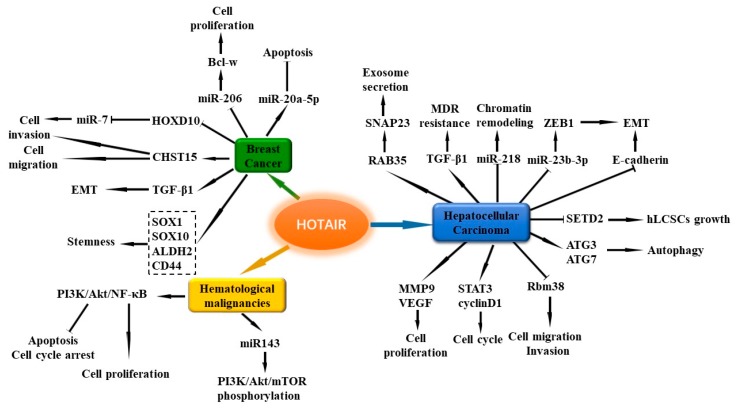
Proposed mechanisms of HOTAIR in various cancers.

**Table 1 cells-08-01015-t001:** Methodologies for exploring the expression, distribution, and function of lncRNAs.

Research Purpose	Available Assays
Function prediction Expression	GIC, CPC2, CNCI, CPAT, ESTScan, PLEK, PORTRAIT, FEELnc, etc. Quantitative real-time PCR (qRT-PCR), Northern blot, gene expression microarray
Location	RNA fluorescent in situ hybridization (FISH), single molecule RNA FISH
Proliferation	3-(4,5-dimethyl-2-thiazolyl)-2,5-diphenyl-2-H-tetrazolium bromide, Thiazolyl Blue Tetrazolium Bromide (MTT) assay, Cell Counting Kit-8 assay, colony formation assay
Apoptosis	Annexin V-fluorescein isothiocyanate (FITC)/propidium iodide (PI) assay, flow cytometry
Cell cycle	Flow cytometry, 5-ethynyl-2′-deoxyuridine (EdU) staining, PI staining, Western blot
Migration and Invasion	Scratch test, Transwell assay
EMT	qRT-PCR, immunohistochemistry, Western blot
Interaction	Luciferase reporter assay, chromatin immunoprecipitation (ChIP), RNA pull-down, RNA-binding protein immunoprecipitation (RIP)
Stemness	Cancer stem cells (CSCs) phenotype assay, self-renewal capacity assay

**Table 2 cells-08-01015-t002:** Summarization of the roles and mechanisms of discussed lncRNAs in the pathogenesis of cancers.

LncRNA	Chromosome Location	Change in Cancer Tissue and Circulation	Roles in the Pathogenesis of Cancers
MALAT1	11q13	↑ in NSCLC tissues ↓in LC circulation ↑in circulation of LC with metastasis	promoting proliferation, differentiation, cell cycle, migration, invasion, EMT, chemoresistance, vasculature formation, neurite outgrowth; inhibiting DNA damage, apoptosis, autophagy;
H19	11p15.5	↑ in AML-M5, CLL, MM, MCL tissues ↑ in LC tissues ↑ in NSCLC circulation ↑ in ER-positive BC tissues ↑ in circulation of BC with ER-positive, PR-positive and LNM	enhancing proliferation, differentiation, migration, invasion, cell cycle, EMT and MET, chemoresistance; suppressing apoptosis; regulating CSCs characteristics;
TUG1	22q12	↓ in LSCC and LAD tissues ↑ in SCLC tissues ↑ in LAD circulation	regulating proliferation, cell cycle, apoptosis, migration, invasion, chemoresistance;
MEG3	14q32.2	↓ in NSCLC tissues ↓ in AML, CML tissues ↓ in NB tissues	inhibiting viability, proliferation, cell cycle, autophagy, chemoresistance; inducing apoptosis;
AFAP1-AS1	4p16.1	↑ in NSCLC tissues	enhancing proliferation, migration, invasion; suppressing apoptosis;
HOTAIR	12q13.13	↑ in BC tissues ↑ in BC circulation ↑ in HCC tissues ↑ in DLBCL, AML, advanced CML tissues	promoting viability, proliferation, cell cycle, migration, invasion, autophagy, EMT, hLCSCs growth, stemness, exosome secretion, chemoresistance; inhibiting apoptosis;
GAS5	1q25.1	↓ in BC tissues ↓ in BC circulation	inhibiting proliferation, migration, invasion, chemoresistance; promoting apoptosis;
UCA1	19p13.12	↑ in BC tissues ↑ in TNBC circulation ↑ in CRC tissues ↑ in CRC circulation	enhancing proliferation, migration, invasion, cell cycle, autophagy, EMT, drug resistance; inhibiting apoptosis;
LincRNA-ROR	18q21.31	↑ in BC tissues ↑ in BC circulation	promoting proliferation, invasion, metastasis, EMT, chemoresistance; regulating CSCs population and self-renewal capacity;
HULC	6p24.3	↑ in HCC tissues ↑ in HCC circulation	promoting proliferation, migration, invasion; inhibiting apoptosis;
NEAT1	11q13.1	↑ in HCC tissues ↑ in HCC circulation	enhancing proliferation, migration, invasion, EMT, drug resistance; suppressing apoptosis;
SNHG1 SNHG7	11q12.3 9q34.3	↑ in HCC tissues ↑ in HCC circulation ↑ in NB tissues	promoting proliferation, invasion and migration, drug resistance, neuronal toxicity; inhibiting apoptosis; promoting proliferation, invasion, migration, EMT, cell cycle arrest;
LincRNA-p21	17	↓ in HCC tissues ↓ in circulation of HCC with metastasis	suppressing proliferation, migration, invasion, EMT, chemoresistance; enhancing apoptosis;
CCAT1	8q24.21	↑ in CRC tissues ↑ in CRC circulation	enhancing proliferation, migration, invasion, EMT; inhibiting cell cycle;
CCAT2	8q24.21	↑ in CRC tissues ↑ in CRC circulation	promoting proliferation, differentiation, migration, invasion, chromosomal instability;
CRNDE	16	↑ in CRC tissues	enhancing proliferation, migration, invasion, chemoresistance; suppressing apoptosis;
PVT1 DLEU1 & DLEU2	8q24.2 13q14.3	↑ in AML-M3, MM tissues ↑ in APL circulation ↓ in CLL, MM tissues	promoting proliferation, cell cycle, telomerase activity; inhibiting apoptosis; inhibiting proliferation, cell cycle, chemoresistance; inducing apoptosis; regulating histone modifications, DNA methylation.

All the detailed information about the change, roles, and mechanisms of the lncRNAs in the table refer to the content in the results section. The symbol “↑” represents upregulation, and the symbol “↓” represents downregulation in cancer tissues and circulation. (Abbreviations: NSCLC, non-small cell lung cancer; LC, lung cancer; CLL, chronic lymphocytic leukemia; MM, multiple myeloma; MCL, mantle cell lymphoma; BC, breast cancer; DLBCL, diffuse large B cell lymphoma; AML, acute myeloid leukemia; CML, chronic myeloid leukemia; TNBC, triple-negative breast cancer; CRC, colorectal cancer; HCC, hepatocellular carcinoma; NB, neuroblastoma; APL, acute promyelocytic leukemia; EMT, epithelial-to-mesenchymal transition; CSC, cancer stem cell; MET, mesenchymal-epithelial transition; hLCSCs, human liver cancer stem cells.)

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
