# Peer review of "Long Non-Coding RNA in the Pathogenesis of Cancers"

_cells, 2019, doi:10.3390/cells8091015_

Round 1

Reviewer 1 Report

The Review Article by Chi and co-authors „Long non-coding RNA in the pathogenesis of cancers” is a long analysis of the literature describing the possible action of lncRNAs in several cancer types. By structuring the manuscript in chapters dedicated to cancer types, several lncRNAs are described more than once making the manuscript very long to read. Given that most of the lncRNAs described seems to be involved in the pathogenesis of more than one cancer type, maybe the authors could dedicate the chapters to lncRNAs and describe their role in more than one cancer type.

Generally the Authors should be careful with the use of abbreviations: the abbreviated names should be written in full the first time are mentioned in the manuscript, later can be only abbreviated. Sometimes the authors use abbreviation for some cancer types and not for others: they should try to be consistent through all the manuscript.

Other points are listed below.

1)      The first part of the Abstract needs to be edited: what does it mean: “With the insidious of cancer, most of the cancers could not be effectively diagnosed at the early stage”? There are also several repetition of the same concept.

2)      On line 53 the authors state that “Most of the lncRNAs are localized in the nucleus”. I would not stress this point as it seems that the picture is much more complex as initially was supposed to be (see for instance ref.: 10.1186/s13059-015-0586-4)

3)      Between lines 70 to 82 the authors discuss about the role of lncRNAs in regulating gene expression through the modulation of chromatin remodeling. Here they write about SWI/SNF and a lncRNA binding to SWI/SNF. The way it is written seems that this is the main mechanism of lncRNAs action, although several other chromatin modifying proteins able to interact with lncRNAs have been described. It needs to be clear that what the authors describe is only an example.

4)      On line 127 and later the authors write about “EMT process”. In my opinion the world process can be omitted.

5)      On line 128 MALTA1 should be MALAT1

6)      On line 133 is not clear the world “downstream” to what refers.

7)      On line 139: I would not call a xenograft a “mouse model”.

8)      On lines 142-144: This sentence is a repetition.

9)      On line 147: remove “in 1990”

10)   On line 208: “also bind to an oncogene HOXB7 promoter” should be: “also bind to the HOXB7 promoter”

11)   On line 244: do the authors mean: transcribed from the AFAP1 promoter?

12)   On line 249: “High expression of AFAP1-AS1 has lymph node metastasis” should be: High expression of AFAP1-AS1 induces lymph node metastasis” or “High expression of AFAP1-AS1 correlates with lymph node metastasis”

13)   On line 261: “to epigenetically down-regulates p21 expression” is “to epigenetically down-regulate p21 expression”

14)   On line 292: if E2 stimulates HOTAIR expression, why is “a target gene of ER-mediated transcriptional repression”?

15)   On line 302: it seems as it is written that the Bcl2 family of proteins plays a crucial role in cellular migration, while this family of proteins a typically functioning in the induction/protection from apoptosis.

16)   On line 311: “and is originally isolated from” should be “and was originally isolated from”.

17)   On line 331: do not start a paragraph with “Moreover”

Author Response

Dear editors and reviewers,

We are hereby submitting our revised manuscript entitled “Long non-coding RNA in the pathogenesis of cancers” for consideration for publication in Cells.

We thank the editors and reviewers for their critical and constructive comments. As detailed below, the manuscript had been thoroughly revised in response to the experts’ comments, and all the changes and new content had been marked in red in the manuscript. Our responses to specific comments of each reviewer are detailed below.   

We are hoping that you will find the revised manuscript will now be suitable for publication in Cells. Thank you for your consideration of this revised manuscript.

Sincerely yours,

Jichun Yang, Ph.D.

Professor of Department of Physiology and Pathophysiology

Center for Non-coding RNA Medicine

Peking University Health Science Center

Reviewer 1

The Review Article by Chi and co-authors „Long non-coding RNA in the pathogenesis of cancers” is a long analysis of the literature describing the possible action of lncRNAs in several cancer types. By structuring the manuscript in chapters dedicated to cancer types, several lncRNAs are described more than once making the manuscript very long to read. Given that most of the lncRNAs described seems to be involved in the pathogenesis of more than one cancer type, maybe the authors could dedicate the chapters to lncRNAs and describe their role in more than one cancer type.

ResponseWe greatly appreciate this important suggestion. At the time to prepare for this reviews, we had considered the idea that dedicate the chapters to lncRNAs. However, in order to discuss the critical lncRNAs in different cancer, we summarized the latest finding regarding the roles and mechanism of some important lncRNAs in five high incidence of cancers (hematological malignancies was added as suggested by the second reviewer). For selecting target lncRNAs, those with significant changes in both cancer tissues and circulating system would be stressed because the literatures about them are relatively comprehensive. Moreover, these kind of lncRNAs are also potential new biomarkers and targets for the diagnosis and treatment of cancers. In addition, we also tried to avoid introducing the same lncRNAs in too many cancers. As suggested by the exert, to make the manuscript easy to understand, the main roles and functions of these lncRNAs in tumorigenesis, metastasis and drug resistance had been revised and summarized in table 2 (the original table 1). Moreover, the main mechanism of two important discussed lncRNAs (MALAT1 and HOTAIR) which plays critical roles in more than two cancers had also been summarized in figures 1 and 2. Meanwhile, the merits and demerits of the discussed lncRNA had also been briefly discussed at the end of each section.

Generally the Authors should be careful with the use of abbreviations: the abbreviated names should be written in full the first time are mentioned in the manuscript, later can be only abbreviated. Sometimes the authors use abbreviation for some cancer types and not for others: they should try to be consistent through all the manuscript.

Response: We thank the expert for this very important suggestion. As suggested, we had carefully read through the whole text, and modified the improper abbreviations. However, the abbreviation of bladder cancer (BC) is the same as breast cancer (BC), thus we did not use the abbreviation BC for bladder cancer. Please see in line 165.

Other points are listed below.

1) The first part of the Abstract needs to be edited: what does it mean: “With the insidious of cancer, most of the cancers could not be effectively diagnosed at the early stage”? There are also several repetition of the same concept.

Response: Thank you for your advice. We have replaced this sentence as: “With the characteristic of insidious onset of cancer, most of the cancers could not be effectively diagnosed at the early stage” in the abstract, please see in lines 15-16.

2) On line 53 the authors state that “Most of the lncRNAs are localized in the nucleus”. I would not stress this point as it seems that the picture is much more complex as initially was supposed to be (see for instance ref.: 10.1186/s13059-015-0586-4)

Response: Thank you for raising this question. Like proteins, the functions of lncRNAs also depend on their subcellular localization. Many lncRNAs are recognized as important modulators for nuclear functions and exhibit distinct nuclear localization patterns (such as HOTAIR, MALAT1, NEAT1). However, some lncRNAs are located in cytoplasm to exert their regulatory roles (such as linc-MD1, NKILA, NORAD, lincRNA-p21). This may be described in this review: Chen et al, Trends Biochem. Sci. 2016. As suggested, we had revised the description accordingly.

    As the reference you mentioned, RNA fluorescence in situ hybridization (RNA FISH) is an approach to study the localization of lncRNA, and lncRNA XIST, MALAT1, NEAT1, and MIAT (Gomafu) which are detected by RNA FISH to show that they localized to nuclear bodies, and the GAS5 which shuttles between the nucleus and cytoplasm. However, the conventional RNA FISH techniques have relatively low sensitivity. In this literature, authors used single molecule RNA FISH in single cells to characterize the sub-cellular localization patterns and abundance of 61 lncRNAs across three human cell types. Using this highly quantitative image-based dataset, they observed a variety of subcellular localization patterns, ranging from bright sub-nuclear foci to almost exclusively cytoplasmic localization. The probe was labeled with a single fluorophore at its 3′ end. When these probes hybridize to a single RNA molecule, the concentration of so many fluorophores at a single location renders the RNA molecule detectable by fluorescence microscopy and showed with a bright point in the picture. We have cited this literature to this sentence, please see in line 61.

3) Between lines 70 to 82 the authors discuss about the role of lncRNAs in regulating gene expression through the modulation of chromatin remodeling. Here they write about SWI/SNF and a lncRNA binding to SWI/SNF. The way it is written seems that this is the main mechanism of lncRNAs action, although several other chromatin modifying proteins able to interact with lncRNAs have been described. It needs to be clear that what the authors describe is only an example.

Response: Thank you for raising this important issue. lncRNAs also modulate chromatin remodeling by regulating the methylation and acetylation of DNA and histone with other molecules, other than SWI/SNF. We had revised these sentences by adding some new content citing new references. Please see in lines 87-88 and 91-94.

4)  On line 127 and later the authors write about “EMT process”. In my opinion the world process can be omitted.

Response: Thank you for your helpful suggestion. We have deleted the word “process”. Please see in line 140.

5)  On line 128 MALTA1 should be MALAT1.

Response: We are sorry for our spelling mistake and we have corrected this word. Please see in line 141.

6)  On line 133 is not clear the world “downstream” to what refers.

Response: Thank you for raising this issue. This sentence means that C-X-C motif chemokine ligand 5 (CXCL5) is a downstream gene of MALAT1, knockdown CXCL5 could inhibit MALAT1-induced cell migration and invasion in vitro. To avoid the misunderstanding, we have replaced this sentence as “C-X-C motif chemokine ligand 5 (CXCL5) is a downstream gene of MALAT1, and knockdown CXCL5 could inhibit MALAT1-induced cell migration and invasion in vitro”. Please see in lines 145-147.

7)  On line 139: I would not call a xenograft a “mouse model”.

Response: Thank you for your helpful suggestion. After tail vein injection of MALAT1 WT or KO cells, the formation of lung tumor nodules was analyzed in nude mice. To avoid the misunderstanding, we have replaced this sentence as “In nude mice, the total number and area of lung tumor nodules are significantly reduced in the injection of A549 MALAT1 KO cells compared to A549 MALAT1 WT cells” in the revised revision. Please see in lines 150-152.

8)  On lines 142-144: This sentence is a repetition.

Response: Thank you for raising this question. This sentence is a conclusion of the role of MALAT1 in this chapter. We have altered some description in our revised manuscript. Please see in lines 155-159.

9)  On line 147: remove “in 1990”

Response: Thanks for you careful reviewing our manuscript. We have removed “in 1990” in the revised revision. Please see in line 162.

10) On line 208: “also bind to an oncogene HOXB7 promoter” should be: “also bind to the HOXB7 promoter”

Response: Thank you for your helpful suggestion. We have replaced this sentence as your advice in your revised manuscript. Please see in line 222.

11) On line 244: do the authors mean: transcribed from the AFAP1 promoter?

Response: Thank you for you arising this issue. AFAP1-AS1 is transcribed from AFAP1 gene in the antisense direction, containing several overlapping and complementary regions among the exons of AFAP1-AS1 and AFAP1. We have added the transcriptional region of AFAP1-AS1 in our revised revision. Please see in lines 259-261.

12) On line 249: “High expression of AFAP1-AS1 has lymph node metastasis” should be: High expression of AFAP1-AS1 induces lymph node metastasis” or “High expression of AFAP1-AS1 correlates with lymph node metastasis”

Response: Thank you for your helpful suggestion. We have replaced the sentence as “High expression of AFAP1-AS1 correlates with lymph node metastasis” in our revised revision. Please see in line 265.

13) On line 261: “to epigenetically down-regulates p21 expression” is “to epigenetically down-regulate p21 expression”

Response: Thanks for you careful reviewing our manuscript. We have corrected this sentence in our revised manuscript. Please see in line 277.

14) On line 292: if E2 stimulates HOTAIR expression, why is “a target gene of ER-mediated transcriptional repression”?

Response: Thank you for raising this question. In estrogen-receptor positive human breast cancer cell MCF-7, Bhan et al (J Mol Biol, 2013) found that the promoter region of HOTAIR contains multiple functional estrogen response elements (EREs). Estrogen receptors (ERs) along with various ER coregulators such as histone methylases MLL1 (mixed lineage leukemia 1) and MLL3 and CREB-binding protein/p300 bind to the promoter of HOTAIR in an E2-dependent manner. Knockdown of ERs and MLLs downregulated the E2-induced HOTAIR expression. It is indicated that HOTAIR is an estrogen-responsive gene. In this study, authors used the concentration of E2 0.1nM significantly increased the expression of HOTAIR. However, the authors also found that high dose of E2 (1nM) inhibited the expression of HOTAIR.          

In another study, Xue et al (Oncogene, 2016) found that high dose of E2 (1nM and 10nM) also suppressed HOTAIR expression, and E2 inhibited HOTAIR expression in a dose- and time-dependent manner in MCF-7 cells. Subsequent studies involving estrogen all tested the concentration of 1nM or 10nM. Then the authors found that HOTAIR is directly repressed by E2 through the ER, and indicated that HOTAIR is a direct target of ER-mediated transcriptional repression. Additional studies discovered that HOTAIR could directly interact with ER, enhance ER transcriptional activities, drive estrogen-independent ER transcriptional program and promote tamoxifen-resistant breast cancer progression.

To avoid the misunderstanding, we have replaced the sentence as: “Another study found that high concentration of E2 could inhibit HOTAIR expression and HOTAIR might act as a target gene of ER-mediated transcriptional repression, additional studies indicate that HOTAIR is also involved in tamoxifen resistance of BC”. Please see in lines 307-309.

15) On line 302: it seems as it is written that the Bcl2 family of proteins plays a crucial role in cellular migration, while this family of proteins a typically functioning in the induction/protection from apoptosis.

Response: Thank you for raising this question. As you mentioned, Bcl2 family of proteins are typically functioning in the induction/protection from apoptosis, such as Bcl2, Bax, Bad et al. However, Bcl2 family include many proteins, which also have roles in regulating cellular migration, especially Bcl-w. In our manuscript, we specifically indicate Bcl-w in this part. Please see in lines 318-321.

16) On line 311: “and is originally isolated from” should be “and was originally isolated from”.

Response: We are sorry for our mistake. We have corrected this sentence in our revised manuscript. Please see in line 341.

17) On line 331: do not start a paragraph with “Moreover”

Response: Thank you for your helpful suggestion. We have deleted “Moreover” in the revised revision. Please see in line 360.

References:

Chen, L.L. Linking Long Noncoding RNA Localization and Function. Trends Biochem. Sci. 2016, 41, 761–772. doi:10.1016/j.tibs.2016.07.003. Bhan, A.; Hussain, I.; Ansari, K.I.; Kasiri, S.; Bashyal, A.; Mandal, S.S. Antisense transcript long noncoding RNA (lncRNA) HOTAIR is transcriptionally induced by estradiol. J Mol Biol 2013, 425, 3707-3722, doi:10.1016/j.jmb.2013.01.022. Xue, X.; Yang, Y.A.; Zhang, A.; Fong, K.W.; Kim, J.; Song, B.; Li, S.; Zhao, J.C.; Yu, J. LncRNA HOTAIR enhances ER signaling and confers tamoxifen resistance in breast cancer. Oncogene 2016, 35, 2746-2755, doi:10.1038/onc.2015.340.

Reviewer 2 Report

In the present manuscript entitled “Long non-coding RNA in the pathogenesis of cancers” by Yujing Chi and colleagues reviewed the literature on the role of Long non-coding RNAs(lncRNAs) in the development of various cancer types. This is an interesting review article. However, this article is too short and did not present a comprehensive review of lncRNAs. A large volume of literature available in the documentation and therefore should be rewritten. 1) A table on the available assays of lncRNAs missing. 2) None of the information related to hematological malignancies is given. 3) Authors should also present the merits and demerits of the discussed literature. 4) Too much text portion. Some pictures depicting the mechanistic aspects of lncRNAs should be given. 5) lncRNAs involvement in the regulation of genes through post-translational modifications is missing. 6) Also, the role in cancer stemness, metastasis, and therapy resistance. 7) Available lncRNA online prediction tools also missing. 8) Biomarkers aspects should be presented. A separate table too. 9) Information related to how to target lncRNAs should also be discussed.

Author Response

Dear editors and reviewers,

We are hereby submitting our revised manuscript entitled “Long non-coding RNA in the pathogenesis of cancers” for consideration for publication in Cells.

We thank the editors and reviewers for their critical and constructive comments. As detailed below, the manuscript had been thoroughly revised in response to the experts’ comments, and all the changes and new content had been marked in red in the manuscript. Our responses to specific comments of each reviewer are detailed below.   

We are hoping that you will find the revised manuscript will now be suitable for publication in Cells. Thank you for your consideration of this revised manuscript.

Sincerely yours,

Jichun Yang, Ph.D.

Professor of Department of Physiology and Pathophysiology

Center for Non-coding RNA Medicine

Peking University Health Science Center

Reviewer 2

In the present manuscript entitled “Long non-coding RNA in the pathogenesis of cancers” by Yujing Chi and colleagues reviewed the literature on the role of Long non-coding RNAs (lncRNAs) in the development of various cancer types. This is an interesting review article. However, this article is too short and did not present a comprehensive review of lncRNAs. A large volume of literature available in the documentation and therefore should be rewritten.

A table on the available assays of lncRNAs

Response: Thank you for raising this question. As suggested, we had added a separated table (Table 1) summarizing the current available assays of lncRNAs in the introduction in our revised revision.

None of the information related to hematological malignancies is given.

Response: Thank you for this great suggestion. As requested, we had added a section discussing the roles of some important lncRNAs in the pathogenesis of hematological malignancies.

Authors should also present the merits and demerits of the discussed literature.

Response: Thank you for raising this question. As suggested, we had briefly discussed the merits and demerits of the discussed lncRNA at the end of the corresponding section.

Too much text portion. Some pictures depicting the mechanistic aspects of lncRNAs should be given.

Response: Thank you for your advice. As suggested, the main roles and mechanisms of two important lncRNAs which play important roles in the pathogenesis of various cancers had been summarized in figures 1 and 2. Moreover, the description of original table 1 (now table 2) had been revised to make it concisely display the key roles and functions of each discussed lncRNAs in the pathogenesis of cancers.

lncRNAs involvement in the regulation of genes through post-translational modifications is missing.

Response: Thank you for this important question. In the post-translational level, lncRNAs mainly regulate protein degradation, protein phosphorylation and protein complex modulation. Some of the discussed lncRNAs did not have the correlated studies. As suggested, we had added some findings regarding the post-translational modifications of genes by lncRNAs in our revised revision.

Also, the role in cancer stemness, metastasis, and therapy resistance.

Response: Thank you for your helpful advice. Some of the lncRNAs did not have the correlated studies regarding cancer stemness. As suggested, we had added some mechanism of lncRNAs in cancer stemness (HOTAIR in BC and HCC, and H19 and LincRNA-ROR in BC). The most of lncRNAs we introduced in the manuscript play crucial roles in cancer metastasis, and therapy resistance. We had described the related mechanisms in detail.

Available lncRNA online prediction tools also missing.

Response: Thank you for raising this very important issue. LncRNAs can be predicted in online prediction tool based on Coding Potential Calculator algorithm version 2 (CPC2) freely at http://cpc2.cbi.pku.edu.cn [1], and also can be predicted using software such as CNCI, CPAT, ESTScan, PLEK, PORTRAIT, FEELnc, TransDecoder and GeneMarkS-T, and CPAT and ESTScan also provide a Web server [2]. Moreover, we had recently developed an effective method, which had been named Gene Importance Calculator (GIC), for predicting the essentiality of lncRNAs [3]. We have added this part in introduction in our revised revision.

Biomarkers aspects should be presente A separate table too.

Response: Thank you for your helpful suggestion. In our previous manuscript conclusion and perspective part, we mentioned that the lncRNAs detected in circulation can be called as biomarker, and we have listed if a lncRNA changes in circulation in cancer in original table 1 (now table 2).

Information related to how to target lncRNAs should also be discussed.

Response: Thanks for you carefully read our manuscript. We have discussed how to target lncRNAs in the part of conclusion and perspective “In particular, several lncRNAs plays important roles in the pathogenesis of various cancers, suggesting that they represent a new class of targets for treating different cancers (Figure 1-2, Table 2). Novel methodologies for treating cancers can be developed by targeting lncRNAs in the future. Potential strategies include targeting their upstream activators or repressors, or directly targeting lncRNA sequences by siRNA.”.

References:

Kang, Y.J.; Yang, D.C.; Kong, L.; Hou, M.; Meng, Y.Q.; Wei, L.; Gao, G. CPC2: a

fast and accurate coding potential calculator based on sequence intrinsic features. Nucleic Acids Res 2017, 45, W12-W16, doi:10.1093/nar/gkx428.

Antonov, I.V.; Mazurov, E.; Borodovsky, M.; Medvedeva, Y.A. Prediction of lncRNAs and their interactions with nucleic acids: benchmarking bioinformatics tools. Brief Bioinform 2019, 20, 551-564, doi:10.1093/bib/bby032. Zeng, P.; Chen, J.; Meng, Y.; Zhou, Y.; Yang, J.; Cui, Q. Defining Essentiality Score of Protein-Coding Genes and Long Noncoding RNAs. Front Genet 2018, 9, 380, doi:10.3389/fgene.2018.00380.

Round 2

Reviewer 1 Report

I don't have any further comments.

Reviewer 2 Report

In the current manuscript entitled “Long non-coding RNA in the pathogenesis of cancers” by Yujing Chi and colleagues have reviewed the current literature on the role of long non-coding RNAs in the pathogenesis of various cancers. I think I have reviewed this article before and now authors revised the manuscript based on the earlier comments. My only concerns are: (1) Lanes 51-60: Corresponding weblinks are missing for the lncRNAs bioinformatic tools; (2) Figures 1 and 2: arrow are too big, should be trimmed; (3) Authors should also add lncRNAs information related to neuroblastoma.

Author Response

Response to revewer

In the current manuscript entitled “Long non-coding RNA in the pathogenesis of cancers” by Yujing Chi and colleagues have reviewed the current literature on the role of long non-coding RNAs in the pathogenesis of various cancers. I think I have reviewed this article before and now authors revised the manuscript based on the earlier comments. My only concerns are:

Lanes 51-60: Corresponding weblinks are missing for the lncRNAs bioinformatic tools;

Response: We thank the expert for raising this very important issue. As suggetesed, we had added the corresponding weblinks in our revised revision based on original references.

Figures 1 and 2: arrow are too big, should be trimmed;

Response: Thank you for this helpful advice. We have changed the shape of the arrow in Figure 1 and 2 in the revised manuscript as suggested.

(3) Authors should also add lncRNAs information related to neuroblastoma.

Response: We greatly thank the expert for this great suggestion. As suggested, we had added a new section discussing and summarizing the roles of some important lncRNAs in the pathogenesis of neuroblastoma (NB).